# Mask-based Latent Reconstruction for Reinforcement Learning

**Tao Yu**[1][*][†]  **Zhizheng Zhang**[2][*]  **Cuiling Lan**[2]  **Yan Lu**[2]  **Zhibo Chen**[1]

[1]University of Science and Technology of China   [2]Microsoft Research Asia

yutao666@mail.ustc.edu.cn, {zhizzhang,culan,yanlu}@microsoft.com
chenzhibo@ustc.edu.cn

## Abstract

For deep reinforcement learning (RL) from pixels, learning effective state representations is crucial for achieving high performance. However, in practice, limited experience and high-dimensional inputs prevent effective representation learning. To address this, motivated by the success of mask-based modeling in other research fields, we introduce mask-based reconstruction to promote state representation learning in RL. Specifically, we propose a simple yet effective self-supervised method, Mask-based Latent Reconstruction (MLR), to predict complete state representations in the latent space from the observations with spatially and temporally masked pixels. MLR enables better use of context information when learning state representations to make them more informative, which facilitates the training of RL agents. Extensive experiments show that our MLR significantly improves the sample efficiency in RL and outperforms the state-of-the-art sample-efficient RL methods on multiple continuous and discrete control benchmarks. Our code is available at https://github.com/microsoft/Mask-based-Latent-Reconstruction.

## 1   Introduction

Learning effective state representations is crucial for reinforcement learning (RL) from visual signals (where a sequence of images is usually the input of an RL agent), such as in DeepMind Control Suite [43], Atari games [5], *etc*. Inspired by the success of mask-based pretraining in the fields of natural language processing (NLP) [11, 37, 38, 7] and computer vision (CV) [4, 21, 47], we make the first endeavor to explore the idea of mask-based modeling in RL.

Mask-based pretraining exploits the reconstruction of masked word embeddings or pixels to promote feature learning in NLP or CV fields. This is, in fact, not straightforwardly applicable for RL due to the following two reasons. First, RL agents learn policies from the interactions with environments, where the experienced states vary as the policy network is updated. Intuitively, collecting additional rollouts for pretraining is often costly especially in real-world applications. Besides, it is challenging to learn effective state representations without the awareness of the learned policy. Second, visual signals usually have high information densities, which may contain distractions and redundancies for policy learning. Thus, for RL, performing reconstruction in the original (pixel) space is not as necessary as it is in the CV or NLP fields.

Based on the analysis above, we study the mask-based modeling tailored to vision-based RL. We present Mask-based Latent Reconstruction (MLR), a simple yet effective self-supervised method, to better learn state representations in RL. Contrary to treating mask-based modeling as a pretraining task in the fields of CV and NLP, our proposed MLR is an auxiliary objective optimized together

---

[*]Equal contribution.
[†]This work was done when Tao Yu was an intern at Microsoft Research Asia.

36th Conference on Neural Information Processing Systems (NeurIPS 2022).

with the policy learning objectives. In this way, the coordination between representation learning and policy learning is considered within a joint training framework. Apart from this, another key difference compared to vision/language research is that we reconstruct masked pixels in the latent space instead of the input space. We take the state representations (*i.e.*, features) inferred from original unmasked frames as the reconstruction targets. This effectively reduces unnecessary reconstruction relative to the pixel-level one and further facilitates the coordination between representation learning and policy learning because the state representations are directly optimized.

Consecutive frames are highly correlated. In MLR, we exploit this property to enable the learned state representations to be more informative, predictive and consistent over both spatial and temporal dimensions. Specifically, we randomly mask a portion of space-time cubes in the input observation (*i.e.*, video clip) sequence and reconstruct the feature representations of the missing contents in the latent space. In this way, similar to the spatial reconstruction for images in [21, 47], MLR enhances the awareness of the agents on the global context information of the entire input observations and promotes the state representations to be predictive in both spatial and temporal dimensions. The global predictive information is encouraged to be encoded into each frame-level state representation, achieving better representation learning and further facilitating policy learning.

We not only propose an effective mask-based modeling method, but also conduct a systematical empirical study for the practices of masking and reconstruction that are as applicable to RL as possible. First, we study the influence of masking strategies by comparing spatial masking, temporal masking and spatial-temporal masking. Second, we investigate the differences between masking and reconstructing in the pixel space and in the latent space. Finally, we study how to effectively add reconstruction supervisions in the latent space.

Our contributions are summarized below:

- We introduce the idea of enhancing representation learning by mask-based modeling to RL to improve the sample efficiency. We integrate the mask-based reconstruction into the training of RL as an auxiliary objective, obviating the need for collecting additional rollouts for pretraining and helping the coordination between representation learning and policy learning in RL.
- We propose Mask-based Latent Reconstruction (MLR), a self-supervised mask-based modeling method to improve the state representations for RL. Tailored to RL, we propose to randomly mask space-time cubes in the pixel space and reconstruct the information of missing contents *in the latent space*. This is shown to be effective for improving the sample efficiency on multiple continuous and discrete control benchmarks (*e.g.*, DeepMind Control Suite [43] and Atari games [5]).
- A systematical empirical study is conducted to investigate the good practices of masking and reconstructing operations in MLR for RL. This demonstrates the effectiveness of our proposed designs in the proposed MLR.

## 2   Related Work

### 2.1   Representation Learning for RL

Reinforcement learning from visual signals is of high practical value in real-world applications such as robotics, video game AI, *etc*. However, such high-dimensional observations may contain distractions or redundant information, imposing considerable challenges for RL agents to learn effective representations [41]. Many prior works address this challenge by taking advantage of self-supervised learning to promote the representation learning of the states in RL. A popular approach is to jointly learn policy learning objectives and auxiliary objectives such as pixel reconstruction [41, 50], reward prediction [23, 41], bisimulation [53], dynamics prediction [41, 13, 28, 29, 39, 52], and contrastive learning of instance discrimination [27] or (spatial -) temporal discrimination [35, 2, 42, 54, 33]. In this line, BYOL [12]-style auxiliary objectives, which are often adopted with data augmentation, show promising performance [39, 52, 48, 19, 14]. More detailed introduction for BYOL and the BYOL-style objectives can be found in Appendix A.3. Another feasible way of acquiring good representations is to pretrain the state encoder to learn effective state representations for the original observations before policy learning. It requires additional offline sample collection or early access to the environments [20, 42, 32, 31, 40], which is not fully consistent with the principle of sample efficiency in practice. This work aims to design a more effective auxiliary task to improve the learned representations toward sample-efficient RL.

## 2.2 Sample-efficient Reinforcement Learning

Collecting rollouts from the interaction with the environment is commonly costly, especially in the real world, leaving the sample efficiency of RL algorithms concerned. To improve the sample efficiency of vision-based RL (*i.e.*, RL from pixel observations), recent works design auxiliary tasks to explicitly improve the learned representations [50, 27–29, 54, 30, 39, 51, 52] or adopt data augmentation techniques, such as random crop/shift, to improve the diversity of data used for training [49, 26]. Besides, there are some model-based methods that learn (world) models in the pixel [55] or latent space [16–18, 51], and perform planning, imagination or policy learning based on the learned models. We focus on the auxiliary task line in this work.

## 2.3 Masked Language/Image Modeling

Masked language modeling (MLM) [11] and its autoregressive variants [37, 38, 7] achieve significant success in the NLP field and produce impacts in other domains. MLM masks a portion of word tokens from the input sentence and trains the model to predict the masked tokens, which has been demonstrated to be generally effective in learning language representations for various downstream tasks. For computer vision (CV) tasks, similar to MLM, masked image modeling (MIM) learns representations for images/videos by pretraining the neural networks to reconstruct masked pixels from visible ones. As an early exploration, Context Encoder [36] apply this idea to Convolutional Neural Network (CNN) model to train a CNN model with a masked region inpainting task. With the recent popularity of the Transformer-based architectures, a series of works [9, 4, 21, 47, 46] dust off the idea of MIM and show impressive performance on learning representations for vision tasks. Inspired by MLM and MIM, we explore the mask-based modeling for RL to exploit the high correlation in vision data to improve agents' awareness of global-scope dynamics in learning state representations. Most importantly, *we propose to predict the masked information in the latent space*, instead of the pixel space like the aforementioned MIM works, which better coordinates the representation learning and the policy learning in RL.

# 3 Approach

## 3.1 Background

Vision-based RL aims to learn policies from pixel observations by interacting with the environment. The learning process corresponds to a partially observable Markov decision process (POMDP) [6, 24], formulated as $(\mathcal{O}, \mathcal{A}, p, r, \gamma)$, where $\mathcal{O}$, $\mathcal{A}$, $p$, $r$ and $\gamma$ denote the observation space (*e.g.*, pixels), the action space, the transition dynamics $p = Pr(\mathbf{o}_{t+1}|\mathbf{o}_{\leq t}, \mathbf{a}_t)$, the reward function $\mathcal{O} \times \mathcal{A} \rightarrow \mathbb{R}$, and the discount factor, respectively. Following common practices [34], an observation $\mathbf{o}_t$ is constructed by a few RGB frames. The transition dynamics and the reward function can be formulated as $p_t = Pr(\mathbf{o}_{t+1}|\mathbf{o}_t, \mathbf{a}_t)$ and $r_t = r(\mathbf{o}_t, \mathbf{a}_t)$, respectively. The objective of RL is to learn a policy $\pi(\mathbf{a}_t|\mathbf{o}_t)$ that maximizes the cumulative discounted return $\mathbb{E}_\pi \sum_{t=0}^{\infty} \gamma^t r_t$, where $\gamma \in [0, 1]$.

## 3.2 Mask-based Latent Reconstruction

Mask-based Latent Reconstruction (MLR) is an auxiliary objective to promote representation learning in vision-based RL and is generally applicable for different RL algorithms, *e.g.*, Soft Actor-Critic (SAC) and Rainbow [22]. The core idea of MLR is to facilitate state representation learning by reconstructing spatially and temporally masked pixels in the latent space. This mechanism enables better use of context information when learning state representations, further enhancing the understanding of RL agents for visual signals. We illustrate the overall framework of MLR in Figure 1 and elaborate on it below.

**Framework.** In MLR, as shown in Figure 1, we mask a portion of pixels in the input observation sequence along its spatial and temporal dimensions. We encode the masked sequence and the original sequence from observations to latent states with an encoder and a momentum encoder, respectively. We perform predictive reconstruction from the states corresponding to the masked sequence while taking the states encoded from the original sequence as the target. We add reconstruction supervisions between the prediction results and the targets in the decoded latent space. The processes of *masking*, *encoding*, *decoding* and *reconstruction* are introduced in detail below.

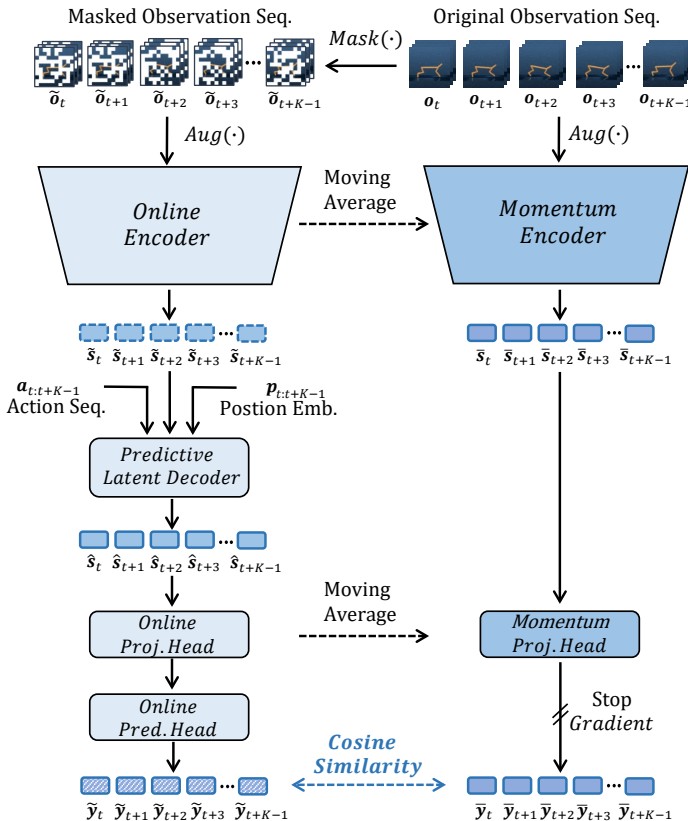

Masked Observation Seq.        Original Observation Seq.

$Mask(\cdot)$

$\tilde{o}_t$ $\tilde{o}_{t+1}$ $\tilde{o}_{t+2}$ $\tilde{o}_{t+3}$ $\tilde{o}_{t+K-1}$     $o_t$ $o_{t+1}$ $o_{t+2}$ $o_{t+3}$ $o_{t+K-1}$

$Aug(\cdot)$                    $Aug(\cdot)$

Online Encoder    Moving Average    Momentum Encoder

$\tilde{s}_t$ $\tilde{s}_{t+1}$ $\tilde{s}_{t+2}$ $\tilde{s}_{t+3}$ $\tilde{s}_{t+K-1}$     $\bar{s}_t$ $\bar{s}_{t+1}$ $\bar{s}_{t+2}$ $\bar{s}_{t+3}$ $\bar{s}_{t+K-1}$

$a_{t:t+K-1}$ Action Seq.        $p_{t:t+K-1}$ Postion Emb.

Predictive Latent Decoder

$\hat{s}_t$ $\hat{s}_{t+1}$ $\hat{s}_{t+2}$ $\hat{s}_{t+3}$ $\hat{s}_{t+K-1}$

Online Proj.Head    Moving Average    Momentum Proj.Head

Online Pred.Head    Cosine Similarity    Stop Gradient

$\tilde{y}_t$ $\tilde{y}_{t+1}$ $\tilde{y}_{t+2}$ $\tilde{y}_{t+3}$ $\tilde{y}_{t+K-1}$     $\bar{y}_t$ $\bar{y}_{t+1}$ $\bar{y}_{t+2}$ $\bar{y}_{t+3}$ $\bar{y}_{t+K-1}$

Figure 1: The framework of the proposed MLR. We perform a random spatial-temporal masking (*i.e.*, *cube* masking) on the sequence of consecutive observations in the pixel space. The masked observations are encoded to be the latent states through an online encoder. We further introduce a predictive latent decoder to decode/predict the latent states conditioned on the corresponding action sequence and temporal positional embeddings. Our method trains the networks to reconstruct the feature representations of the missing contents in an appropriate *latent* space using a cosine similarity based distance metric applied between the predicted features of the reconstructed states and the target features inferred from original observations by momentum networks.

**(i) Masking.** Given an observation sequence of $K$ timesteps $\tau_K^o = \{o_t, o_{t+1}, \cdots, o_{t+K-1}\}$ and each observation containing $n$ RGB frames, all the observations in the sequence are stacked to be a cuboid with the shape of $K \times H \times W$ where $H \times W$ is the spatial size. Note that the actual shape is $K \times H \times W \times D$ where $D = 3n$ is the number of channels in each observation. We omit the channel dimension for simplicity. As illustrated in Figure 2, we divide the cuboid into regular non-overlapping *cubes* with the shape of $k \times h \times w$. We then randomly mask a portion of the cubes following a uniform distribution and obtain a masked observation sequence $\tilde{\tau}_K^o = \{\tilde{o}_t, \tilde{o}_{t+1}, \cdots, \tilde{o}_{t+K-1}\}$. Following [52], we perform stochastic image augmentation $Aug(\cdot)$ (*e.g.*, random crop and in-tensity) on each masked observation in $\tilde{\tau}_K^o$. The objective of MLR is to predict the state representations of the unmasked observation sequence from the masked one in the latent space.

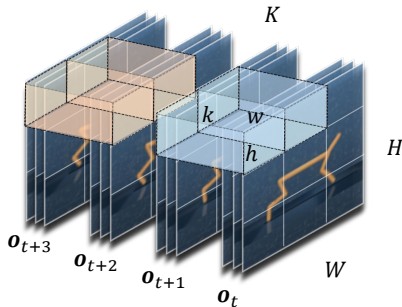

Figure 2: Illustration of our cube masking. We divide the input observation sequence to non-overlapping cubes ($k \times h \times w$). In this example, we have $k = \frac{1}{2}K$, $h = \frac{1}{3}H$ and $w = \frac{1}{3}W$ where the observation sequence has $K = 4$ timesteps and a spatial size of $H \times W$.

**(ii) Encoding.** We adopt two encoders to learn state representations from masked observations and original observations respectively. A CNN-based encoder $f$ is used to encode each masked observation $\tilde{\mathbf{o}}_{t+i}$ into its corresponding latent state $\tilde{\mathbf{s}}_{t+i} \in \mathbb{R}^d$. After the encoding, we obtain a sequence of the masked latent states $\tilde{\tau}_K^s = \{\tilde{\mathbf{s}}_t, \tilde{\mathbf{s}}_{t+1}, \cdots, \tilde{\mathbf{s}}_{t+K-1}\}$ for masked observations. The parameters of this encoder are updated based on gradient back-propagation in an end-to-end way. We thus call it "online" encoder. The state representations inferred from original observations are taken as the targets of subsequently described reconstruction. To make them more robust, inspired by [27, 39, 52], we exploit another encoder for the encoding of original observations. This encoder, called "momentum" encoder as in Figure 1, has the same architecture as the online encoder, and its parameters are updated by an exponential moving average (EMA) of the online encoder weights $\theta_f$ with the momentum coefficient $m \in [0, 1)$, as formulated below:

$$\bar{\theta}_f \leftarrow m\bar{\theta}_f + (1 - m)\theta_f. \tag{1}$$

**(iii) Decoding.** Similar to the mask-based modeling in the CV field, *e.g.*, [21, 47], the online encoder in our MLR predicts the information of the masked contents from the unmasked contents. As opposed to pixel-space reconstruction in [21, 47], MLR performs the reconstruction in the latent space to better perform RL policy learning. Moreover, pixels usually have high information densities [21] which may contain distractions and redundancies for the policy learning in RL. Considering the fact that in RL the next state is determined by the current state as well as the action, we propose to utilize both the actions and states as the inputs for prediction to reduce the possible ambiguity to enable an RL-tailored mask-based modeling design. To this end, we adopt a transformer-based latent decoder to infer the reconstruction results via a global message passing where the actions and temporal information are exploited. Through this process, the predicted information is "passed" to its corresponding state representations.

As shown in Figure 3, the input tokens of the latent decoder consist of both the masked state sequence $\tilde{\tau}_K^s$ (*i.e.*, state tokens) and the corresponding action sequence $\tau_K^a = \{\mathbf{a}_t, \mathbf{a}_{t+1}, \cdots, \mathbf{a}_{t+K-1}\}$ (*i.e.*, action tokens). Each action token is embedded as a feature vector with the same dimension as the stated token using an embedding layer, through an embedding layer $Emb(\cdot)$. Following the common practices in transformer-based models [45], we adopt the relative positional embeddings to encode relative temporal positional information into both state and action tokens with an element-wise addition denoted by "+" in the following equation (see Appendix B.1 for more details). Notably, the state and action token at the same timestep $t + i$ share the same positional embedding $\mathbf{p}_{t+i} \in \mathbb{R}^d$. Thus, the inputs of latent decoder can be mathematically represented as follows:

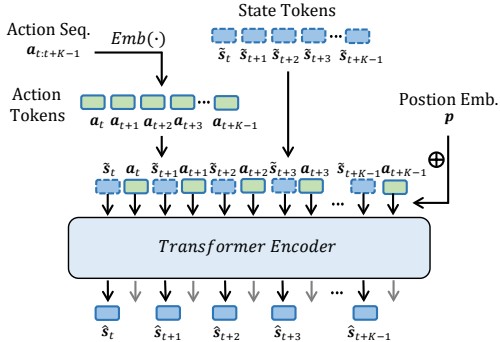

Figure 3: Illustration of predictive latent decoder.

$$\mathbf{x} = [\tilde{\mathbf{s}}_t, \mathbf{a}_t, \tilde{\mathbf{s}}_{t+1}, \mathbf{a}_{t+1}, \cdots, \tilde{\mathbf{s}}_{t+K-1}, \mathbf{a}_{t+K-1}] + [\mathbf{p}_t, \mathbf{p}_t, \mathbf{p}_{t+1}, \mathbf{p}_{t+1}, \cdots, \mathbf{p}_{t+K-1}, \mathbf{p}_{t+K-1}]. \tag{2}$$

The input token sequence is fed into a Transformer encoder [45] consisting of $L$ attention layers. Each layer is composed of a Multi-Headed Self-Attention (MSA) layer [45], a layer normalisation (LN) [3], and multilayer perceptron (MLP) blocks. The process can be described as follows:

$$\mathbf{z}^l = \text{MSA}(\text{LN}(\mathbf{x}^l)) + \mathbf{x}^l, \tag{3}$$

$$\mathbf{x}^{l+1} = \text{MLP}(\text{LN}(\mathbf{z}^l)) + \mathbf{z}^l. \tag{4}$$

The output tokens of the latent decoder, *i.e.*, $\hat{\tau}_K^s = \{\hat{\mathbf{s}}_t, \hat{\mathbf{s}}_{t+1}, \cdots, \hat{\mathbf{s}}_{t+K-1}\}$, are the predictive reconstruction results for the latent representations inferred from the original observations. We elaborate on the reconstruction loss between the prediction results and the corresponding targets in the following.

**(iv) Reconstruction.** Motivated by the success of BYOL [12] in self-supervised learning, we use an asymmetric architecture for calculating the distance between the predicted/reconstructed latent states and the target states, similar to [39, 52]. For the outputs of the latent decoder, we use a projection

head $g$ and a prediction head $q$ to get the final prediction result $\hat{\mathbf{y}}_{t+i} = q(g(\hat{\mathbf{s}}_{t+i}))$ corresponding to $\mathbf{s}_{t+i}$. For the encoded results of original observations, we use a momentum-updated projection head $\bar{g}$ whose weights are updated with an EMA of the weights of the online projection head. These two projection heads have the same architectures. The outputs of the momentum projection head $\bar{g}$, $i.e.$, $\bar{\mathbf{y}}_{t+i} = \bar{g}(\bar{\mathbf{s}}_{t+i})$, are the final reconstruction targets. Here, we apply a stop-gradient operation as illustrated in Figure 1 to avoid model collapse, following [12].

The objective of MLR is to enforce the final prediction result $\hat{\mathbf{y}}_{t+i}$ to be as close as possible to its corresponding target $\bar{\mathbf{y}}_{t+i}$. To achieve this, we design the reconstruction loss in our proposed MLR by calculating the cosine similarity between $\hat{\mathbf{y}}_{t+i}$ and $\bar{\mathbf{y}}_{t+i}$, which can be formulated below:

$$\mathcal{L}_{mlr} = 1 - \frac{1}{K} \sum_{i=0}^{K-1} \frac{\hat{\mathbf{y}}_{t+i}}{\|\hat{\mathbf{y}}_{t+i}\|_2} \frac{\bar{\mathbf{y}}_{t+i}}{\|\bar{\mathbf{y}}_{t+i}\|_2}. \tag{5}$$

The loss $\mathcal{L}_{mlr}$ is used to update the parameters of the online networks, including encoder $f$, predictive latent decoder $\phi$, projection head $g$ and prediction head $q$. Through our proposed self-supervised auxiliary objective MLR, the learned state representations by the encoder will be more informative and thus can further facilitate policy learning.

**Objective.** The proposed MLR serves as an auxiliary task, which is optimized together with the policy learning. Thus, the overall loss function $\mathcal{L}_{total}$ for RL agent training is:

$$\mathcal{L}_{total} = \mathcal{L}_{rl} + \lambda \mathcal{L}_{mlr}, \tag{6}$$

where $\mathcal{L}_{rl}$ and $\mathcal{L}_{mlr}$ are the loss functions of the base RL agent (*e.g.*, SAC [15] and Rainbow [22]) and the proposed mask-based latent reconstruction, respectively. $\lambda$ is a hyperparameter for balancing the two terms. Notably, the agent of vision-based RL commonly consists of two parts, *i.e.*, the (state) representation network (*i.e.*, encoder) and the policy learning network. The encoder of MLR is taken as the representation network to encode observations into the state representations for RL training. The latent decoder can be discarded during testing since it is only needed for the optimization with our proposed auxiliary objective during training. More details can be found in Appendix B.

## 4 Experiment

### 4.1 Setup

**Environments and Evaluation.** We evaluate the sample efficiency of our MLR on both the continuous control benchmark DeepMind Control Suite (DMControl) [43] and the discrete control benchmark Atari [5]. On DMControl, following the previous works [27, 49, 26, 52], we choose six commonly used environments from DMControl, *i.e.*, *Finger, spin*; *Cartpole, swingup*; *Reacher, easy*; *Cheetah, run*; *Walker, walk* and *Ball in cup, catch* for the evaluation. We test the performance of RL agents with the mean and median scores over 10 episodes at 100k and 500k environment steps, referred to as **DMControl-100k** and **DMControl-500k** benchmarks, respectively. The score of each environment ranges from 0 to 1000 [43]. For discrete control, we test agents on the **Atari-100k** benchmark [55, 44, 25, 27] which contains 26 Atari games and allows the agents 100k interaction steps (*i.e.*, 400k environment steps with action repeat of 4) for training. Human-normalized score (HNS) [3] is used to measure the performance on each game. Considering the high variance of the scores on this benchmark, we test each run over 100 episodes [39]. To achieve a more rigorous evaluation on high-variance benchmarks such as Atari-100k with a few runs, recent work Rliable [1] systematically studies the evaluation bias issue in deep RL and recommends robust and efficient aggregate metrics to evaluate the overall performance (across all tasks and runs), *e.g.*, interquartile-mean (IQM) and optimality gap (OG) [4], with percentile confidence intervals (CIs, estimated by the percentile bootstrap with stratified sampling). We follow aforementioned common practices and report the aggregate metrics on the Atari-100k benchmark with 95% CIs.

---

[3]HNS is calculated by $\frac{S_A - S_R}{S_H - S_R}$, where $S_A$, $S_R$ and $S_H$ are the scores of the agent, random play and the expert human, respectively.

[4]IQM discards the top and bottom 25% of the runs and calculates the mean score of the remaining 50% runs. OG is the amount by which the agent fails to meet a minimum score of the default human-level performance. Higher IQM and lower OG are better.

Table 1: Comparison results (mean ± std) on the DMControl-100k and DMControl-500k benchmarks. Our method augments *Baseline* with the proposed MLR objective (denoted as *MLR*).

| 100k Step Scores | PlaNet | Dreamer | SAC+AE | SLAC | CURL | DrQ | PlayVirtual | Baseline | MLR |
|---|---|---|---|---|---|---|---|---|---|
| Finger, spin | $136 \pm 216$ | $341 \pm 70$ | $740 \pm 64$ | $693 \pm 141$ | $767 \pm 56$ | $901 \pm 104$ | $\mathbf{915 \pm 49}$ | $853 \pm 112$ | $907 \pm 58$ |
| Cartpole, swingup | $297 \pm 39$ | $326 \pm 27$ | $311 \pm 11$ | - | $582 \pm 146$ | $759 \pm 92$ | $\mathbf{816 \pm 36}$ | $784 \pm 63$ | $806 \pm 48$ |
| Reacher, easy | $20 \pm 50$ | $314 \pm 155$ | $274 \pm 14$ | - | $538 \pm 233$ | $601 \pm 213$ | $785 \pm 142$ | $593 \pm 118$ | $\mathbf{866 \pm 103}$ |
| Cheetah, run | $138 \pm 88$ | $235 \pm 137$ | $267 \pm 24$ | $319 \pm 56$ | $299 \pm 48$ | $344 \pm 67$ | $474 \pm 50$ | $399 \pm 80$ | $\mathbf{482 \pm 38}$ |
| Walker, walk | $224 \pm 48$ | $277 \pm 12$ | $394 \pm 22$ | $361 \pm 73$ | $403 \pm 24$ | $612 \pm 164$ | $460 \pm 173$ | $424 \pm 281$ | $\mathbf{643 \pm 114}$ |
| Ball in cup, catch | $0 \pm 0$ | $246 \pm 174$ | $391 \pm 82$ | $512 \pm 110$ | $769 \pm 43$ | $913 \pm 53$ | $926 \pm 31$ | $648 \pm 287$ | $\mathbf{933 \pm 16}$ |
| Mean | 135.8 | 289.8 | 396.2 | 471.3 | 559.7 | 688.3 | 729.3 | 616.8 | **772.8** |
| Median | 137.0 | 295.5 | 351.0 | 436.5 | 560.0 | 685.5 | 800.5 | 620.5 | **836.0** |
| **500k Step Scores** | | | | | | | | | |
| Finger, spin | $561 \pm 284$ | $796 \pm 183$ | $884 \pm 128$ | $673 \pm 92$ | $926 \pm 45$ | $938 \pm 103$ | $963 \pm 40$ | $944 \pm 97$ | $\mathbf{973 \pm 31}$ |
| Cartpole, swingup | $475 \pm 71$ | $762 \pm 27$ | $735 \pm 63$ | - | $841 \pm 45$ | $868 \pm 10$ | $865 \pm 11$ | $871 \pm 4$ | $\mathbf{872 \pm 5}$ |
| Reacher, easy | $210 \pm 390$ | $793 \pm 164$ | $627 \pm 58$ | - | $929 \pm 44$ | $942 \pm 71$ | $942 \pm 66$ | $943 \pm 52$ | $\mathbf{957 \pm 41}$ |
| Cheetah, run | $305 \pm 131$ | $570 \pm 253$ | $550 \pm 34$ | $640 \pm 19$ | $518 \pm 28$ | $660 \pm 96$ | $719 \pm 51$ | $602 \pm 67$ | $674 \pm 37$ |
| Walker, walk | $351 \pm 58$ | $897 \pm 49$ | $847 \pm 48$ | $842 \pm 51$ | $902 \pm 43$ | $921 \pm 4575$ | $928 \pm 30$ | $818 \pm 263$ | $\mathbf{939 \pm 10}$ |
| Ball in cup, catch | $460 \pm 380$ | $879 \pm 87$ | $794 \pm 58$ | $852 \pm 71$ | $959 \pm 27$ | $963 \pm 9$ | $\mathbf{967 \pm 5}$ | $960 \pm 10$ | $964 \pm 14$ |
| Mean | 393.7 | 782.8 | 739.5 | 751.8 | 845.8 | 882.0 | **897.3** | 856.3 | 896.5 |
| Median | 405.5 | 794.5 | 764.5 | 757.5 | 914.0 | 929.5 | 935.0 | 907.0 | **948.0** |

**Implementation.** SAC [15] and Rainbow [22] are taken as the base continuous-control agent and discrete-control agent, respectively (see Appendix A.1 and A.2 for the details of the two algorithms). In our experiments, we denote the base agents trained only by RL loss $\mathcal{L}_{rl}$ (as in Equation 6) as *Baseline*, while denoting the models of applying our proposed MLR to the base agents as *MLR* for the brevity. Note that compared to naive SAC or Rainbow, our *Baseline* additionally adopts data augmentation (random crop and random intensity). We adopt this following the prior works [26, 49] which uncovers that applying proper data augmentation can significantly improve the sample efficiency of SAC or Rainbow. As shown in Equation 6, we set a weight $\lambda$ to balance $\mathcal{L}_{rl}$ and $\mathcal{L}_{mlr}$ so that the gradients of these two loss items lie in a similar range and empirically find $\lambda = 1$ works well in most environments. In MLR, by default, we set the length of a sampled trajectory $K$ to 16 and mask ratio $\eta$ to 50%. We set the size of the masked cube ($k \times h \times w$) to $8 \times 10 \times 10$ on most DMControl tasks and $8 \times 12 \times 12$ on the Atari games. More implementation details can be found in Appendix B.

## 4.2 Comparison with State-of-the-Arts

**DMControl.** We compare our proposed MLR with the state-of-the-art (SOTA) sample-efficient RL methods proposed for continuous control, including PlaNet [16], Dreamer [17], SAC+AE [50], SLAC [28], CURL [27], DrQ [49] and PlayVirtual [52]. The comparison results are shown in Table 1, and all results are averaged over 10 runs with different random seeds. We can observe that: (i) MLR significantly improves *Baseline* in both DMControl-100k and DMControl-500k, and achieves consistent gains relative to *Baseline* across all environments. It is worthy to mention that, for the DMControl-100k, our proposed method outperforms *Baseline* by **25.3%** and **34.7%** in mean and median scores, respectively. This demonstrates the superiority of MLR in improving the sample efficiency of RL algorithms. (ii) The RL agent equipped with MLR outperforms most state-of-the-art methods on DMControl-100k and DMControl-500k. Specifically, our method surpasses the best previous method (*i.e.*, PlayVirtual) by 43.5 and 35.5 respectively on mean and median scores on DMControl-100k. Our method also delivers the best median score and reaches a comparable mean score with the strongest SOTA method on DMControl-500k.

**Atari-100k.** We further compare MLR with the SOTA model-free methods for discrete control, including DER [44], OTR [25], CURL [27], DrQ [49], DrQ($\epsilon$) (DrQ using the $\epsilon$-greedy parameters in [8]) , SPR [39] and PlayVirtual [52]. These methods and our MLR for Atari are all based on Rainbow [22]. The Atari-100k results are shown in Figure 4. MLR achieves an interquartile-mean (IQM) score of 0.432, which is **28.2%** higher than SPR (IQM: 0.337) and **15.5%** higher than PlayVirtual (IQM: 0.374). This indicates that MLR has the highest sample efficiency overall. For the optimality gap (OG) metric, MLR reaches an OG of 0.522 which is better than SPR (OG: 0.577) and PlayVirtual (OG: 0.558), showing that our method performs closer to the desired target, *i.e.*, human-level performance. The full scores of MLR (over 3 random seeds) across the 26 Atari games and more comparisons and analysis can be found in Appendix C.1.

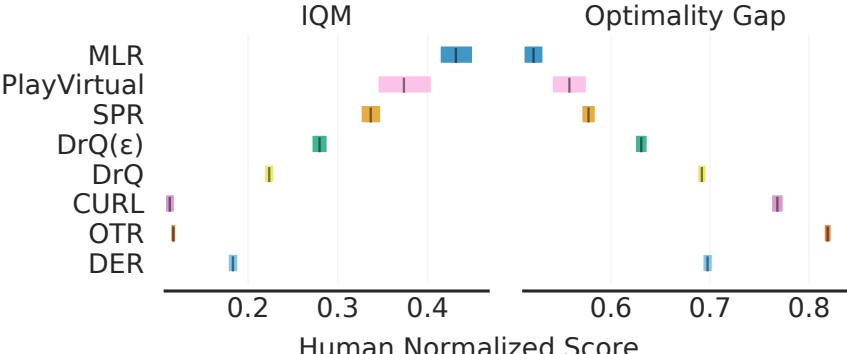

Figure 4: Comparison results on Atari-100k. Aggregate metrics (IQM and optimality gap (OG)) [1] with 95% confidence intervals (CIs) are used for the evaluation. Higher IQM and lower OG are better.

## 4.3  Ablation Study

In this section, we investigate the effectiveness of our MLR auxiliary objective, the impact of masking strategies and the model/method design. Unless otherwise specified, we conduct the ablation studies on the DMControl-100k benchmark and run each model with 5 random seeds.

**Effectiveness evaluation.** Besides the comparison with SOTA methods, we demonstrate the effectiveness of our proposed MLR by studying its improvements compared to our *Baseline*. The numerical results are presented in Table 1, while the curves of test performance during the training process are given in Appendix C.2. Both numerical results and test performance curves can demonstrate that our method obviously outperforms *Baseline* across different environments thanks to more informative representations learned by MLR.

**Masking strategy.** We compare three design choices of the masking operation: (i) *Spatial masking* (denoted as *MLR-S*): we randomly mask *patches* for each frame independently. (ii) *Temporal masking* (denoted as *MLR-T*): we divide the input observation sequence into multiple segments along the temporal dimension and mask out a portion of segments randomly. (Here, the segment length is set to be equal to the temporal length of cube, *i.e.*, $k$.) (iii) *Spatial-temporal masking* (also referred to as "cube masking"): as aforementioned and illustrated in Figure 2, we rasterize the observation sequence into non-overlapping cubes and randomly mask a portion of them. Except for the differences described above, other configurations for masking remain the same as our proposed spatial-temporal (*i.e.*, cube) masking. From the results in Table 2, we have the following observations: (i) All three masking strategies (*i.e.*, *MLR-S*, *MLR-T* and *MLR*) achieve mean score improvements compared to *Baseline* by **18.5%**, **12.2%** and **25.0%**, respectively, and achieve median score improvements by **23.4%**, **25.0%** and **35.9%**, respectively. This demonstrates the effectiveness of the core idea of introducing mask-based reconstruction to improve the representation learning of RL. (ii) Spatial-temporal masking is the most effective strategy over these three design choices. This strategy matches better with the nature of video data due to its spatial-temporal continuity in masking. It encourages the state representations to be more predictive and consistent along the spatial and temporal dimensions, thus facilitating policy learning in RL.

**Reconstruction target.** In masked language/image modeling, reconstruction/prediction is commonly performed in the original signal space, such as word embeddings or pixels. To study the influence of the reconstruction targets for the task of RL, we compare two different reconstruction spaces: (i) *Pixel space reconstruction* (denoted as *MLR-Pixel*): we predict the masked contents directly by reconstructing original pixels, like the practices in CV and NLP domains; (ii) *Latent space reconstruction* (*i.e.*, *MLR*): we reconstruct the state representations (*i.e.*, features) of original observations from masked observations, as we proposed in MLR. Table 2 shows the comparison results. The reconstruction in the latent space is superior to that in the pixel space in improving sample efficiency. As discussed in the preceding sections, vision data might contain distractions and redundancies for policy learning, making the pixel-level reconstruction unnecessary. Besides, latent space reconstruction is more conducive to the coordination between the representation learning and the policy learning in RL, because the state representations are directly optimized.

Table 2: Ablation study of masking strategy and reconstruction target. We compare three masking strategies: spatial masking (*MLR-S*), temporal masking (*MLR-T*) and spatial-temporal masking (*MLR*), and two reconstruction targets: original pixels (denoted as *MLR-Pixel*) and momentum projections in the latent space (*i.e.*, *MLR*).

| Environment | Baseline | MLR-S | MLR-T | MLR-Pixel | MLR |
|---|---|---|---|---|---|
| Finger, spin | $822 \pm 146$ | $\mathbf{919 \pm 55}$ | $787 \pm 139$ | $782 \pm 95$ | $907 \pm 69$ |
| Cartpole, swingup | $782 \pm 74$ | $665 \pm 118$ | $\mathbf{829 \pm 33}$ | $803 \pm 91$ | $791 \pm 50$ |
| Reacher, easy | $557 \pm 137$ | $848 \pm 82$ | $745 \pm 84$ | $787 \pm 136$ | $\mathbf{875 \pm 92}$ |
| Cheetah, run | $438 \pm 33$ | $449 \pm 46$ | $443 \pm 43$ | $346 \pm 84$ | $\mathbf{495 \pm 13}$ |
| Walker, walk | $414 \pm 310$ | $556 \pm 189$ | $393 \pm 202$ | $490 \pm 216$ | $\mathbf{597 \pm 102}$ |
| Ball in cup, catch | $669 \pm 310$ | $927 \pm 6$ | $934 \pm 29$ | $675 \pm 292$ | $\mathbf{939 \pm 9}$ |
| Mean | 613.7 | 727.3 | 688.5 | 647.2 | **767.3** |
| Median | 613.0 | 756.5 | 766.0 | 728.5 | **833.0** |

**Mask ratio.** In recent works of masked image modeling [21, 47], the mask ratio is found to be crucial for the final performance. We study the influences of different masking ratios on sample efficiency in Figure 5 (with 3 random seeds) and find that the ratio of 50% is an appropriate choice for our proposed MLR. An over-small value of this ratio could not eliminate redundancy, making the objective easy to be reached by extrapolation from neighboring contents that are free of capturing and understanding semantics from visual signals. An over-large value leaves few contexts for achieving the reconstruction goal. As discussed in [21, 47], the choice of this ratio varies for different modalities and depends on the information density of the input signals.

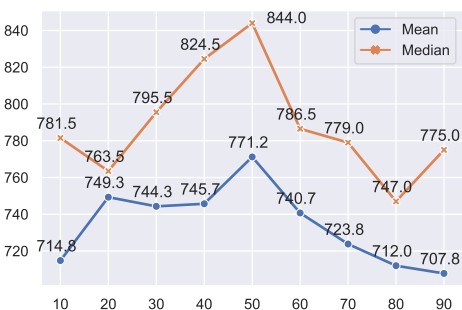

Figure 5: Ablation study of mask ratio.

**Action token.** We study the contributions of action tokens used for the latent decoder in Figure 1 by discarding it from our proposed framework. The results are given in Table 3. Intuitively, prediction only from visual signals is of more or less ambiguity. Exploiting action tokens benefits by reducing such ambiguity so that the gradients of less uncertainty can be obtained for updating the encoder.

Table 3: Ablation studies on action token, masking features and momentum decoder. *MLR w.o. ActTok* denotes removing the action tokens in the input tokens of the predictive latent decoder. *MLR-F* indicates performing masking on convolutuional feature maps. *MLR-MoDec* indicates adding a momentum predictive latent decoder in the target networks.

| Environment | Baseline | MLR w.o. ActTok | MLR-F | MLR-MoDec | MLR |
|---|---|---|---|---|---|
| Finger, spin | $822 \pm 146$ | $832 \pm 46$ | $828 \pm 143$ | $843 \pm 135$ | $\mathbf{907 \pm 69}$ |
| Cartpole, swingup | $782 \pm 74$ | $\mathbf{816 \pm 27}$ | $789 \pm 55$ | $766 \pm 88$ | $791 \pm 50$ |
| Reacher, easy | $557 \pm 137$ | $835 \pm 51$ | $753 \pm 159$ | $800 \pm 49$ | $\mathbf{875 \pm 92}$ |
| Cheetah, run | $438 \pm 33$ | $433 \pm 72$ | $477 \pm 38$ | $470 \pm 12$ | $\mathbf{495 \pm 13}$ |
| Walker, walk | $414 \pm 310$ | $412 \pm 210$ | $\mathbf{673 \pm 33}$ | $571 \pm 152$ | $597 \pm 102$ |
| Ball in cup, catch | $669 \pm 310$ | $837 \pm 114$ | $843 \pm 119$ | $788 \pm 155$ | $\mathbf{939 \pm 9}$ |
| Mean | 613.7 | 694.2 | 727.2 | 706.4 | **767.3** |
| Median | 613.0 | 824.2 | 771.0 | 777.0 | **833.0** |

**Masking features.** We compare "masking pixels" and "masking features" in Table 3. Masking features (denoted by *MLR-F*) does not perform equally well compared with masking pixels as proposed in MLR, but it still achieves significant improvements relative to *Baseline*.

**Why not use a latent decoder for targets?** We have tried to add a momentum-updated latent decoder for obtaining the target states $\bar{s}_{t+i}$ as illustrated in Figure 1. The results in Table 3 show that adding a momentum decoder leads to performance drops. Since there is no information prediction for the state representation learning from the original observation sequence without masking, we do not need to refine the implicitly predicted information like that in the outputs $\hat{s}_{t+i}$ of the online encoder.

**Decoder depth.** We analyze the influence of using Transformer-based latent decoders with different depths. We show the experimental results in Table 4. Generally, deeper latent decoders lead to worse sample efficiency with lower mean and median scores. Similar to the designs in [21, 47], it is appropriate to use a lightweight decoder in MLR, because we expect the predicting masked information to be mainly completed by the encoder instead of the decoder. Note that the state representations inferred by the encoder are adopted for the policy learning in RL.

Table 4: Ablation study of predictive latent decoder depth. We report the number of parameters, mean and median scores on DMControl-100k.

| Layers | Param. | Mean | Median |
|--------|--------|--------|--------|
| 1 | 20.4K | 726.8 | 719.0 |
| 2 | 40.8K | **767.3** | **833.0** |
| 4 | 81.6K | 766.2 | 789.5 |
| 8 | 163.2K | 728.3 | 763.5 |

**Similarity loss.** We compare models using two kinds of similarity metrics to measure the distance in the latent space and observe that using cosine similarity loss is better than using mean squared error (MSE) loss. The results and analysis can be found in Appendix C.2.

**Projection and prediction heads.** Previous works [12, 39, 10] have shown that in self-supervised learning, supervising the feature representations in the projected space via the projection/prediction heads is often better than in the original feature space. We investigate the effect of the two heads and find that both improve agent performance (see Appendix C.2).

**Sequence length and cube size.** These two factors can be viewed as hyperparameters. Their corresponding experimental analysis and results are in Appendix C.2.

## 4.4 More Analysis

We make more detailed investigation and analysis of our MLR from the following aspects with details shown in Appendix C.3. (i) We provide a more detailed analysis of the effect of each masking strategy. (ii) The quality of the learned representations are further evaluated through a pretraining evaluation and a regression accuracy test. (iii) We also investigate the performance of MLR on more challenging control tasks and find that MLR is still effective. (iv) We additionally discuss the relationship between PlayVirtual and MLR. They both achieve leading performance in improving sample efficiency but from different perspectives. (v) We discuss the applications and limitations of MLR. We observe that MLR is more effective on tasks with backgrounds and viewpoints that do not change drastically than on tasks with drastically changing backgrounds/viewpoints or vigorously moving objects.

## 5 Conclusion

In this work, we make the first effort to introduce the mask-based modeling to RL for facilitating policy learning by improving the learned state representations. We propose MLR, a simple yet effective self-supervised auxiliary objective to reconstruct the masked information in the latent space. In this way, the learned state representations are encouraged to include richer and more informative features. Extensive experiments demonstrate the effectiveness of MLR and show that MLR achieves the state-of-the-art performance on both DeepMind Control and Atari benchmarks. We conduct a detailed ablation study for our proposed designs in MLR and analyze their differences from that in NLP and CV domains. We hope our proposed method can further inspire research for vision-based RL from the perspective of improving representation learning. Moreover, the concept of masked latent reconstruction is also worthy of being explored and extended in CV and NLP fields. We are looking forward to seeing more mutual promotion between different research fields.

## Acknowledgments

We thank all the anonymous reviewers for their valuable comments on our paper.

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
