# Mask-based Latent Reconstruction for Reinforcement Learning (Appendix)

**Tao Yu**[1*†] **Zhizheng Zhang**[2*] **Cuiling Lan**[2] **Yan Lu**[2] **Zhibo Chen**[1]
[1]University of Science and Technology of China [2]Microsoft Research Asia
yutao666@mail.ustc.edu.cn, {zhizzhang,culan,yanlu}@microsoft.com
chenzhibo@ustc.edu.cn

## A  Extended Background

### A.1  Soft Actor-Critic

Soft Actor-Critic (SAC) [9] is an off-policy actor-critic algorithm, which is based on the maximum entropy RL framework where the standard return maximization objective is augmented with an entropy maximization term [32]. SAC has a soft Q-function $Q$ and a policy $\pi$. The soft Q-function is learned by minimizing the soft Bellman error:

$$J(Q) = \mathbb{E}_{tr \sim \mathcal{D}}[(Q(\mathbf{s}_t, \mathbf{a}_t) - (r_t + \gamma \bar{V}(\mathbf{s}_{t+1}))^2], \tag{1}$$

where $tr = (\mathbf{s}_t, \mathbf{a}_t, r_t, \mathbf{s}_{t+1})$ is a tuple with current state $\mathbf{s}_t$, action $\mathbf{a}_t$, successor $\mathbf{s}_{t+1}$ and reward $r_t$, $\mathcal{D}$ is the replay buffer and $\bar{V}$ is the target value function. $\bar{V}$ has the following expectation:

$$\bar{V}(\mathbf{s}_t) = \mathbb{E}_{\mathbf{a}_t \sim \pi}[\bar{Q}(\mathbf{s}_t, \mathbf{a}_t) - \alpha \log \pi(\mathbf{a}_t|\mathbf{s}_t)], \tag{2}$$

where $\bar{Q}$ is the target Q-function whose parameters are updated by an exponential moving average of the parameters of the Q-function $Q$, and the temperature $\alpha$ is used to balance the return maximization and the entropy maximization. The policy $\pi$ is represented by using the reparameterization trick and optimized by minimizing the following objective:

$$J(\pi) = \mathbb{E}_{\mathbf{s}_t \sim \mathcal{D}, \epsilon_t \sim \mathcal{N}}[\alpha \log \pi(f_\pi(\epsilon_t; \mathbf{s}_t)|\mathbf{s}_t) - Q(\mathbf{s}_t, f_\pi(\epsilon_t; \mathbf{s}_t))], \tag{3}$$

where $\epsilon_t$ is the input noise vector sampled from Gaussian distribution $\mathcal{N}(0, I)$, and $f_\pi(\epsilon_t; \mathbf{s}_t)$ denotes actions sampled stochastically from the policy $\pi$, *i.e.*, $f_\pi(\epsilon_t; \mathbf{s}_t) \sim \tanh(\mu_\pi(\mathbf{s}_t) + \sigma_\pi(\mathbf{s}_t) \odot \epsilon_t)$. SAC is shown to have a remarkable performance in continuous control [9].

### A.2  Deep Q-network and Rainbow

Deep Q-network (DQN) [17] trains a neural network $Q_\theta$ with parameters $\theta$ to approximate the Q-function. DQN introduces a target Q-network $Q_{\theta'}$ for stable training. The target Q-network $Q_{\theta'}$ has the same architecture as the Q-network $Q_\theta$, and the parameters $\theta'$ are updated with $\theta$ every certain number of iterations. The objective of DQN is to minimize the squared error between the predictions of $Q_\theta$ and the target values provided by $Q_{\theta'}$:

$$J(Q_\theta) = (Q_\theta(\mathbf{s}_t, a_t) - (r_t + \gamma \max_a Q_{\theta'}(\mathbf{s}_{t+1}, a)))^2 \tag{4}$$

Rainbow [12] integrates a number of improvements on the basis of the vanilla DQN [17] including: (i) employing modified target Q-value sampling in Double DQN [24]; (ii) adopting Prioritized Experience Replay [18] strategy; (iii) decoupling the value function of state and the advantage function of action from Q-function like Dueling DQN [27]; (iv) introducing distributional RL and predicting value distribution as C51 [4]; (v) adding parametric noise into the network parameters like NoisyNet [6]; and (vi) using multi-step return [21]. Rainbow is typically regarded as a strong model-free baseline for discrete control.

---

*Equal contribution.
†This work was done when Tao Yu was an intern at Microsoft Research Asia.

36th Conference on Neural Information Processing Systems (NeurIPS 2022).

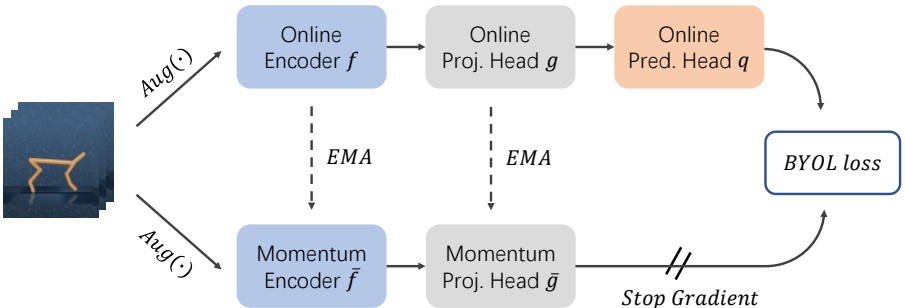

Figure 1: An illustration of the framework of BYOL [7].

### A.3 BYOL-style Auxiliary Objective

BYOL [7] is a strong self-supervised representation learning method by enforcing the similarity of the representations of the same image across diverse data augmentation. The pipeline is shown in Figure 1. BYOL has an online branch and a momentum branch. The momentum branch is used for computing a stable target for learning representations [11, 22]. BYOL is composed of an online encoder $f$, a momentum encoder $\bar{f}$, an online projection head $g$, a momentum projection head $\bar{g}$ and an prediction head $q$. The momentum encoder and projection head have the same architectures as the corresponding online networks and are updated by an exponential moving average (EMA) of the online weights (see Equation 1 in the main manuscript). The prediction head is only used in the online branch, making BYOL's architecture asymmetric. Given an image $x$, BYOL first produces two views $v$ and $v'$ from $x$ through images augmentations. The online branch outputs a representation $y = f(v)$ and a projection $z = g(y)$, and the momentum branch outputs $y' = \bar{f}(v')$ and a target projection $z' = \bar{g}(y')$. BYOL then use a prediction head $q$ to regress $z'$ from $z$, *i.e.*, $q(z) \rightarrow z'$. BYOL minimizes the similarity loss between $q(z)$ and a stop-gradient[3] target $sg(z')$.

$$\mathcal{L}_{BYOL} = \|q(z) - sg(z')\|_2^2 = 2 - 2 \frac{q(z)}{\|q(z)\|_2} \frac{sg(z')}{\|sg(z')\|_2}. \tag{5}$$

Inspired by the success of BYOL in learning visual representations, recent works introduce BYOL-style learning objectives to vision-based RL for learning effective state representations and show promising performance [20, 31, 28, 10, 8]. The BYOL-style learning is often integrated into auxiliary objectives in RL such as future state prediction [20, 8], cycle-consistent dynamics prediction [31], prototypical representation learning [28] and invariant representation learning [10]. These works also show that it is more effective to supervise/regularize the predicted representations in the BYOL's projected latent space than in the representation or pixel space. Besides, the BYOL-style auxiliary objectives are commonly trained with data augmentation since it can conveniently produce two BYOL views. For example, SPR [20] and PlayVirtual [31] apply random crop and random intensity to input observations in Atari games. The proposed MLR auxiliary objective can be categorized into BYOL-style auxiliary objectives.

## B Implementation Detail

### B.1 Network Architecture

Our model has two parts: the basic networks and the auxiliary networks. The basic networks are composed of a representation network (*i.e.*, encoder) $f$ parameterized by $\theta_f$ and the policy learning networks $\omega$ (*e.g.*, SAC [9] or Rainbow [12]) parameterized by $\theta_\omega$.

We follow CURL [16] to build the architecture of the basic networks on the DMControl [23] benchmarks. The encoder is composed of four convolutional layers (with a rectified linear units (ReLU) activation after each), a fully connected (FC) layer, and a layer normalization (LN) [2] layer. Furthermore, the policy learning networks are built by multilayer perceptrons (MLPs). For the basic networks on Atari [3], we also follow CURL to adopt the original architecture of Rainbow [12]

---

[3]Stop gradient operation stops the gradients from passing through, avoiding trivial solutions [11].

where the encoder consists of three convolutional layers (with a ReLU activation after each), and the Q-learning heads are MLPs.

Our auxiliary networks have online networks and momentum (or target) networks. The online networks consist of an encoder $f$, a predictive latent decoder (PLD) $\phi$, a projection head $g$ and a prediction head $q$, parameterized by $\theta_f$, $\theta_\phi$, $\theta_g$ and $\theta_q$, respectively. Notably, the encoders in the basic networks and the auxiliary networks are *shared*. As shown in Figure 1 in our main manuscript, there are a momentum encoder $\bar{f}$ and a momentum projection head $\bar{g}$ for computing the self-supervised targets. The momentum networks have the same architectures as the corresponding online networks. Our PLD is a transformer encoder [26] and has two standard attention layers (with a single attention head). We use an FC layer as the action embedding head to transform the original action into an embedding which has the same dimension as the state representation (*i.e.*, state token). The transformer encoder treats each input token as independent of the other. Thus, positional embeddings, which encode the positional information of tokens in a sequence, are added to the input tokens of PLD to maintain their relative temporal positional information (*i.e.*, the order of the state/action sequences). We use sine and cosine functions to build the positional embeddings following [26]:

$$p_{(pos,2j)} = \sin(pos/10000^{2j/d}), \tag{6}$$

$$p_{(pos,2j+1)} = \cos(pos/10000^{2j/d}), \tag{7}$$

where $pos$ is the position, $j$ is the dimension and $d$ is the embedding size (equal to the state representation size). We follow PlayVirtual [31] in the architecture design of the projection and the prediction heads, built by MLPs and FC layers.

## B.2 Training Detail

**Optimization and training.** The training algorithm of our method is presented in Algorithm 1. We use Adam optimizer [14] to optimize all trainable parameters in our model, with $(\beta_1, \beta_2) = (0.9, 0.999)$ (except for $(0.5, 0.999)$ for SAC temperature $\alpha$). Modest data augmentation such as crop/shift is shown to be effective for improving RL agent performance in vision-based RL [29, 15, 20, 31]. Following [31, 20, 16], we use random crop and random intensity in training the auxiliary objective, *i.e.*, $\mathcal{L}_{mlr}$. Besides, we warmup the learning rate of our MLR objective on DMControl by

$$lr = lr_0 \cdot \min(step\_num^{-0.5}, step\_num \cdot warmup\_step^{-1.5}), \tag{8}$$

where the $lr$ and $lr_0$ denote the current learning rate and the initial learning rate, respectively, and $step\_num$ and $warmup\_step$ denote the current step and the warmup step, respectively. We empirically find that the warmup schedule bring improvements on DMControl.

**Hyperparameters.** We present all hyperparameters used for the DMControl benchmarks [23] in Table 8 and the Atari-100k benchmark in Table 9. We follow prior work [31, 20, 16] for the policy learning hyperparameters (*i.e.*, SAC and Rainbow hyperparameters). The hyperparameters specific to our MLR auxiliary objective, including MLR loss weight $\lambda$, mask ratio $\eta$, the length of the sampled sequence $K$, cube shape $k \times h \times w$ and the depth of the decoder $L$, are shown in the bottom of the tables. By default, we set $\lambda$ to 1, $\eta$ to 50%, $L$ to 2, $k$ to 8, and $h \times w$ to $10 \times 10$ on DMControl and $12 \times 12$ on Atari. We exceptionally set $k$ to 4 in *Cartpole-swingup* and *Reacher-easy* on DMControl due to their large motion range, and $\lambda$ to 5 in *Pong* and *Up N Down* on Atari as their MLR losses are relatively smaller than the rest 24 Atari games.

**Baseline and data augmentation.** Our baseline models (*Baseline*) are equivalent to CURL [16] without the auxiliary loss except for the slight differences on the applied data augmentation strategies. CURL adopts random crop for data augmentation. We follow the prior work SPR [20] and PlayVirtual [31] to adopt random crop and random intensity for both our *Baseline* and MLR.

**GPU and wall-clock time.** In our experiment, we run MLR with a single GPU ( NVIDIA Tesla V100 or GeForce RTX 3090) for each environment. MLR has the same inference time complexity as *Baseline* since both use only the encoder and policy learning head during testing. On Atari, the average wall-clock training time is 6.0, 10.9, 4.0, and 8.2 hours for SPR, PlayVirtual, Baseline and MLR, respectively. On DMControl, the training time is 4.3, 5.2, 3.8, 6.5 hours for SPR, PlayVirtual, Baseline and MLR, respectively. We will leave the optimization of our training strategy as future work to speed up the training, *e.g.*, reducing the frequency of using masked-latent reconstruction. The evaluation is based on a single GeForce RTX 3090 GPU.

---

**Algorithm 1** Training Algorithm for MLR

---

**Require:** An online encoder $f$, a momentum encoder $\bar{f}$, a predictive latent decoder $\phi$, an online projection head $g$, a momentum projection head $\bar{g}$, a prediction head $q$ and policy learning networks $\omega$, parameterized by $\theta_f, \bar{\theta}_f, \theta_\phi, \theta_g, \bar{\theta}_g, \theta_q$ and $\theta_\omega$, respectively; a stochastic cube masking function $Mask(\cdot)$; a stochastic image augmentation function $Aug(\cdot)$; an optimizer $Optimize(\cdot, \cdot)$.

1: Determine auxiliary loss weight $\lambda$, sequence length $K$, mask ratio $\eta$, cube size $k \times h \times w$ and EMA coefficient $m$.
2: Initialize a replay buffer $\mathcal{D}$.
3: Initialize $Mask(\cdot)$ with $\eta$ and $k \times h \times w$.
4: Initialize all network parameters.
5: **while** $train$ **do**
6:     Interact with the environment based on the policy
7:     Collect the transition: $\mathcal{D} \leftarrow \mathcal{D} \cup (\mathbf{o}, \mathbf{a}, \mathbf{o}_{next}, r)$
8:     Sample a trajectory of $K$ timesteps $\{\mathbf{o}_t, \mathbf{a}_t, \mathbf{o}_{t+1}, \mathbf{a}_{t+1}, \cdots, \mathbf{o}_{t+K-1}, \mathbf{a}_{t+K-1}\}$ from $\mathcal{D}$
9:     Initialize losses: $\mathcal{L}_{mlr} \leftarrow 0; \mathcal{L}_{rl} \leftarrow 0$
10:     Randomly mask the observation sequence:
        $\{\tilde{\mathbf{o}}_t, \tilde{\mathbf{o}}_{t+1}, \cdots, \tilde{\mathbf{o}}_{t+K-1}\} \leftarrow Mask(\{\mathbf{o}_t, \mathbf{o}_{t+1}, \cdots, \mathbf{o}_{t+K-1}\})$
11:     Perform augmentation and encoding:
        $\{\tilde{\mathbf{s}}_t, \tilde{\mathbf{s}}_{t+1}, \cdots, \tilde{\mathbf{s}}_{t+K-1}\} \leftarrow \{f(Aug(\tilde{\mathbf{o}}_t)), f(Aug(\tilde{\mathbf{o}}_{t+1})), \cdots, f(Aug(\tilde{\mathbf{o}}_{t+K-1}))\}$
12:     Perform decoding:
        $\{\hat{\mathbf{s}}_t, \hat{\mathbf{s}}_{t+1}, \cdots, \hat{\mathbf{s}}_{t+K-1}\} \leftarrow \phi(\{\tilde{\mathbf{s}}_t, \tilde{\mathbf{s}}_{t+1}, \cdots, \tilde{\mathbf{s}}_{t+K-1}\}; \{\mathbf{a}_t, \mathbf{a}_{t+1}, \cdots, \mathbf{a}_{t+K-1}\})$
13:     Perform projection and prediction:
        $\{\hat{\mathbf{y}}_t, \hat{\mathbf{y}}_{t+1}, \cdots, \hat{\mathbf{y}}_{t+K-1}\} \leftarrow \{q(g(\hat{\mathbf{s}}_t)), q(g(\hat{\mathbf{s}}_{t+1})), \cdots, q(g(\hat{\mathbf{s}}_{t+K-1}))\}$
14:     Calculate targets:
        $\{\bar{\mathbf{y}}_t, \bar{\mathbf{y}}_{t+1}, \cdots, \bar{\mathbf{y}}_{t+K-1}\} \leftarrow \{\bar{g}(\bar{f}(Aug(\mathbf{o}_t))), \bar{g}(\bar{f}(Aug(\mathbf{o}_{t+1}))),$
        $\cdots, \bar{g}(\bar{f}(Aug(\mathbf{o}_{t+K-1})))\}$
15:     Calculate MLR loss: $\mathcal{L}_{mlr} \leftarrow 1 - \frac{1}{K} \sum_{i=0}^{K-1} \frac{\hat{\mathbf{y}}_{t+i}}{\|\hat{\mathbf{y}}_{t+i}\|_2} \frac{\bar{\mathbf{y}}_{t+i}}{\|\bar{\mathbf{y}}_{t+i}\|_2}$
16:     Calculate RL loss $\mathcal{L}_{rl}$ based on a given base RL algorithm (*e.g.*, SAC)
17:     Calculate total loss: $\mathcal{L}_{total} \leftarrow \mathcal{L}_{rl} + \lambda \mathcal{L}_{mlr}$
18:     Update online parameters: $(\theta_f, \theta_\phi, \theta_g, \theta_q, \theta_\omega) \leftarrow Optimize((\theta_f, \theta_\phi, \theta_g, \theta_q, \theta_\omega), \mathcal{L}_{total})$
19:     Update momentum parameters: $(\bar{\theta}_f, \bar{\theta}_g) \leftarrow m(\bar{\theta}_f, \bar{\theta}_g) + (1-m)(\theta_f, \theta_g)$
20: **end while**

---

### B.3 Environment and Code

DMControl [23] and Atari [3] are widely used environment suites in RL community, which are public and do not involve personally identifiable information or offensive contents. We use the two environment suites to evaluate model performance. The implementation of MLR is based on the open-source PlayVirtual [31] codebase [4]. The statistical tools on Atari are obtained from the open-source library *rliable* [5][1].

## C More Experimental Results and Analysis

### C.1 More Atari-100k Results

We present the comparison results across all 26 games on the Atari-100k benchmark in Table 1. Our MLR reaches the highest scores on 11 out of 26 games and outperforms the compared methods on the aggregate metrics, *i.e.*, interquartile-mean (IQM) and optimality gap (OG) with 95% confidence intervals (CIs). Notably, MLR improve the *Baseline* performance by 47.9% on IQM, which shows the effectiveness of our proposed auxiliary objective. We also present the *performance profiles*[6] using

---

[4]Link: https://github.com/microsoft/Playvirtual, licensed under the MIT License.

[5]Link: https://github.com/google-research/rliable, licensed under the Apache License 2.0.

[6]Performance profiles [5] show the tail distribution of scores on combined runs across tasks [1]. Performance profiles of a distribution $X$ is calculated by $\hat{F}_X(\tau) = \frac{1}{M} \sum_{m=1}^{M} \frac{1}{N} \sum_{n=1}^{N} \mathbb{1}[x_{m,n} > \tau]$, indicating the fraction of runs above a score $\tau$ across $N$ tasks and $M$ seeds.

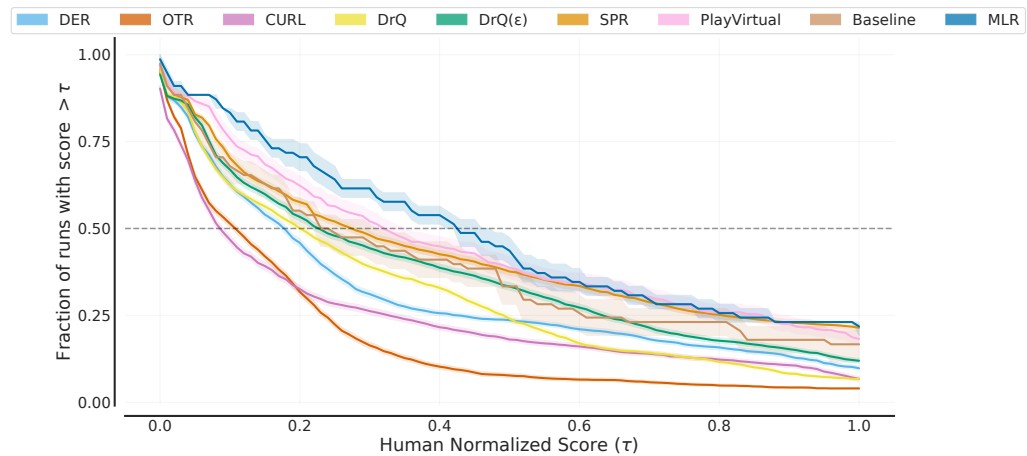

Figure 2: Performance profiles on the Atari-100k benchmark based on human-normalized score distributions. Shaded regions indicates 95% confidence bands. The score distribution of MLR is clearly superior to previous methods and *Baseline*.

human-normalized scores (HNS) with 95% CIs in Figure 2. The performance profiles confirm the superiority and effectiveness of our MLR. The results of DER [25], OTR [13], CURL [16], DrQ [29] and SPR [20] are from *rliable* [1], based on 100 random seeds. The results of PlayVirtual [31] are based on 15 random seeds, and the results of *Baseline* and MLR are averaged over 3 random seeds and each run is evaluated with 100 episodes. We report the standard deviations across runs of *Baseline* and MLR in Table 2.

Table 1: Comparison on the Atari-100k benchmark. Our method reaches the highest scores on 11 out of 26 games and the best performance concerning the aggregate metrics, *i.e.*, interquartile-mean (IQM) and optimality gap (OG) with 95% confidence intervals. Our method augments *Baseline* with the MLR objective and achieves a 47.9% relative improvement on IQM.

| Game | Human | Random | DER | OTR | CURL | DrQ | SPR | PlayVirtual | Baseline | MLR |
|---|---|---|---|---|---|---|---|---|---|---|
| Alien | 7127.7 | 227.8 | 802.3 | 570.8 | 711.0 | 734.1 | 841.9 | 947.8 | 678.5 | **990.1** |
| Amidar | 1719.5 | 5.8 | 125.9 | 77.7 | 113.7 | 94.2 | 179.7 | 165.3 | 132.8 | **227.7** |
| Assault | 742.0 | 222.4 | 561.5 | 330.9 | 500.9 | 479.5 | 565.6 | **702.3** | 493.3 | 643.7 |
| Asterix | 8503.3 | 210.0 | 535.4 | 334.7 | 567.2 | 535.6 | 962.5 | 933.3 | **1021.3** | 883.7 |
| Bank Heist | 753.1 | 14.2 | 185.5 | 55.0 | 65.3 | 153.4 | **345.4** | 245.9 | 288.2 | 180.3 |
| Battle Zone | 37187.5 | 2360.0 | 8977.0 | 5139.4 | 8997.8 | 10563.6 | 14834.1 | 13260.0 | 13076.7 | **16080.0** |
| Boxing | 12.1 | 0.1 | -0.3 | 1.6 | 0.9 | 6.6 | 35.7 | **38.3** | 14.3 | 26.4 |
| Breakout | 30.5 | 1.7 | 9.2 | 8.1 | 2.6 | 15.4 | 19.6 | **20.6** | 16.7 | 16.8 |
| Chopper Cmd | 7387.8 | 811.0 | 925.9 | 813.3 | 783.5 | 792.4 | **946.3** | 922.4 | 878.7 | 910.7 |
| Crazy Climber | 35829.4 | 10780.5 | 34508.6 | 14999.3 | 9154.4 | 21991.6 | **36700.5** | 23176.7 | 28235.7 | 24633.3 |
| Demon Attack | 1971.0 | 152.1 | 627.6 | 681.6 | 646.5 | **1142.4** | 517.6 | 1131.7 | 310.5 | 854.6 |
| Freeway | 29.6 | 0.0 | 20.9 | 11.5 | 28.3 | 17.8 | 19.3 | 16.1 | **30.9** | 30.2 |
| Frostbite | 4334.7 | 65.2 | 871.0 | 224.9 | 1226.5 | 508.1 | 1170.7 | 1984.7 | 994.3 | **2381.1** |
| Gopher | 2412.5 | 257.6 | 467.0 | 539.4 | 400.9 | 618.0 | 660.6 | 684.3 | 650.9 | **822.3** |
| Hero | 30826.4 | 1027.0 | 6226.0 | 5956.6 | 4987.7 | 3722.6 | 5858.6 | **8597.5** | 4661.2 | 7919.3 |
| Jamesbond | 302.8 | 29.0 | 275.7 | 88.0 | 331.0 | 251.8 | 366.5 | 394.7 | 270.0 | **423.2** |
| Kangaroo | 3035.0 | 52.0 | 581.7 | 348.5 | 740.2 | 974.5 | 3617.4 | 2384.7 | 5036.0 | **8516.0** |
| Krull | 2665.5 | 1598.0 | 3256.9 | 3655.9 | 3049.2 | **4131.4** | 3681.6 | 3880.7 | 3571.3 | 3923.1 |
| Kung Fu Master | 22736.3 | 258.5 | 6580.1 | 6659.6 | 8155.6 | 7154.5 | **14783.2** | 14259.0 | 10517.3 | 10652.0 |
| Ms Pacman | 6951.6 | 307.3 | 1187.4 | 908.0 | 1064.0 | 1002.9 | 1318.4 | 1335.4 | 1320.9 | **1481.3** |
| Pong | 14.6 | -20.7 | -9.7 | -2.5 | -18.5 | -14.3 | -5.4 | -3.0 | -3.1 | **4.9** |
| Private Eye | 69571.3 | 24.9 | 72.8 | 59.6 | 81.9 | 24.8 | 86.0 | 93.9 | 93.3 | **100.0** |
| Qbert | 13455.0 | 163.9 | 1773.5 | 552.5 | 727.0 | 934.2 | 866.3 | **3620.1** | 553.8 | 3410.4 |
| Road Runner | 7845.0 | 11.5 | 11843.4 | 2606.4 | 5006.1 | 8724.7 | 12213.1 | **13429.4** | 12337.0 | 12049.7 |
| Seaquest | 42054.7 | 68.4 | 304.6 | 272.9 | 315.2 | 310.5 | 558.1 | 532.9 | 471.9 | **628.3** |
| Up N Down | 11693.2 | 533.4 | 3075.0 | 2331.7 | 2646.4 | 3619.1 | **10859.2** | 10225.2 | 4112.8 | 6675.7 |
| Interquartile Mean | 1.000 | 0.000 | 0.183 | 0.117 | 0.113 | 0.224 | 0.337 | 0.374 | 0.292 | **0.432** |
| Optimality Gap | 0.000 | 1.000 | 0.698 | 0.819 | 0.768 | 0.692 | 0.577 | 0.558 | 0.614 | **0.522** |

Table 2: Standard deviations (STDs) of *Baseline* and MLR on Atari-100k. The STDs are calculated based on 3 random seeds.

| Game | Baseline | MLR | Game | Baseline | MLR | Game | Baseline | MLR |
|---|---|---|---|---|---|---|---|---|
| Alien | 61.2 | 79.2 | Crazy Climber | 12980.1 | 2334.5 | Kung Fu Master | 5026.9 | 971.8 |
| Amidar | 70.7 | 48.0 | Demon Attack | 89.8 | 149.7 | Ms Pacman | 124.3 | 249.4 |
| Assault | 4.3 | 28.0 | Freeway | 0.3 | 1.0 | Pong | 11.5 | 3.1 |
| Asterix | 32.1 | 43.3 | Frostbite | 1295.3 | 607.3 | Private Eye | 11.5 | 0.0 |
| Bank Heist | 39.2 | 49.7 | Gopher | 7.9 | 266.0 | Qbert | 346.6 | 96.3 |
| Battle Zone | 2070.2 | 1139.6 | Hero | 2562.3 | 692.0 | Road Runner | 4632.6 | 669.1 |
| Boxing | 7.3 | 7.4 | Jamesbond | 108.2 | 20.8 | Seaquest | 107.1 | 92.2 |
| Breakout | 1.8 | 1.4 | Kangaroo | 3801.2 | 3025.9 | Up N Down | 919.3 | 531.5 |
| Chopper Command | 119.9 | 242.1 | Krull | 605.9 | 792.0 | | | |

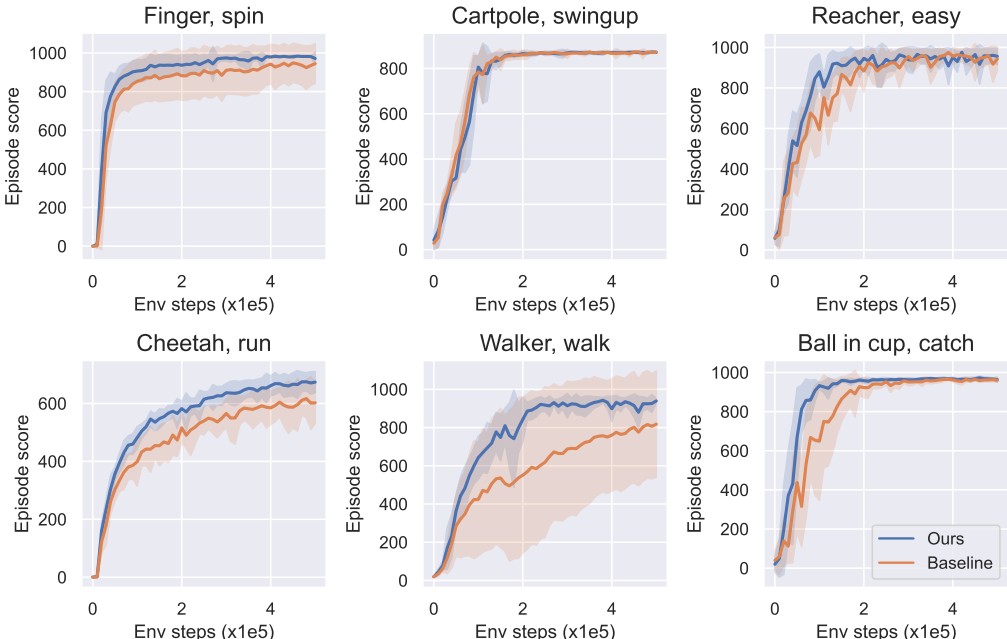

Figure 3: Test performance during the training period (500k environment steps). Lines denote the mean scores over 10 random seeds, and the shadows are the corresponding standard deviations. In most environments on DMControl, our results (blue lines) are consistently better than *Baseline* (orange lines).

## C.2 Extended Ablation Study

We give more details of the ablation study. Again, unless otherwise specified, we conduct the ablations on DMControl-100k with 5 random seeds.

**Effectiveness evaluation.** Besides the numerical results in Table 1 in the main manuscript, we present the test score curves during the training process in Figure 3. Each curve is drawn based on 10 random seeds. The curves demonstrate the effectiveness of the proposed MLR objective. Besides, we observe that MLR achieves more significant gains on the relatively challenging tasks (*e.g.*, *Walker-walk* and *Cheetah-run* with six control dimensions) than on the easy tasks (*e.g.*, *Cartpole-swingup* with a single control dimension). This is because solving challenging tasks often needs more effective representation learning, leaving more room for MLR to play its role.

**Similarity loss.** MLR performs the prediction in the latent space where the value range of the features is unbounded. Using cosine similarity enables the optimization to be conducted in a normalized space, which is more robust to outliers. We compare MLR models using mean squared error (MSE) loss and cosine similarity loss in Table 3. We find that using MSE loss is worse than using cosine similarity loss. Similar observations are found in SPR [20] and BYOL [7].

Table 3: Ablation study of similarity loss (*SimLoss*), projection (*Proj.*) and prediction (*Pred.*) heads.

| Model | SimLoss | Proj. | Pred. | Mean | Median |
|---|---|---|---|---|---|
| Baseline | - | - | - | 613.7 | 613.0 |
| | Cosine | | | 722.5 | 770.0 |
| MLR | Cosine | | ✓ | 750.3 | 819.5 |
| | Cosine | ✓ | ✓ | **767.3** | **833.0** |
| | MSE | ✓ | ✓ | 704.8 | 721.0 |

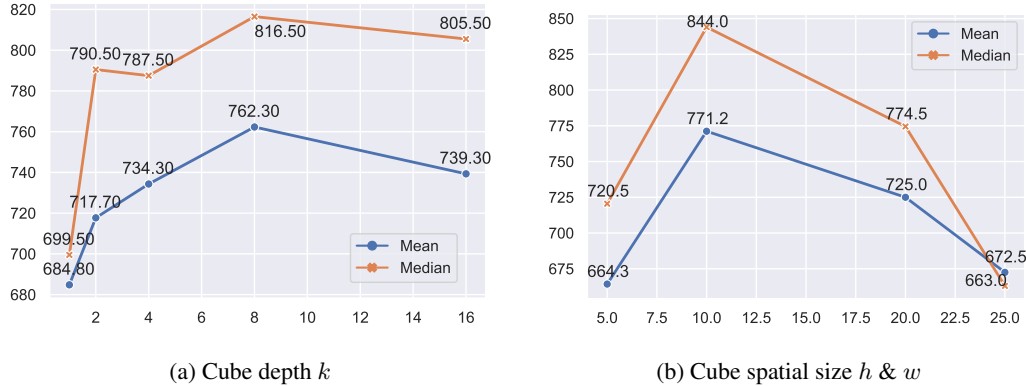

(a) Cube depth $k$      (b) Cube spatial size $h$ & $w$

Figure 4: Ablation studies of (a) cube depth $k$ and (b) cube spatial size $h$ & $w$. The result of each model is averaged over 3 random seeds.

**Projection and prediction heads.** MLR adopt the projection and prediction heads following the widely used design in prior works [7, 20]. We study the impact of the heads in Table 3. The results show that using both projection and prediction heads performs best, which is consistent with the observations in the aforementioned prior works.

**Sequence length.** Table 4 shows the results of the observation sequence length $K$ at {8, 16, 24}. A large $K$ (*e.g.*, 24) does not bring further performance improvement as the network can reconstruct the missing content in a trivial way like copying and pasting the missing content from other states. In contrast, a small $K$ like 8 may not be sufficient for learning rich context information. A sequence length of 16 is a good trade-off in our experiment.

Table 4: Ablation study of sequence length $K$.

| Env. | Baseline | K=8 | K=16 | K=24 |
|---|---|---|---|---|
| Finger, spin | $822 \pm 146$ | $816 \pm 129$ | $\mathbf{907 \pm 69}$ | $875 \pm 63$ |
| Cartpole, swingup | $782 \pm 74$ | $\mathbf{857 \pm 3}$ | $791 \pm 50$ | $781 \pm 58$ |
| Reacher, easy | $557 \pm 137$ | $779 \pm 116$ | $\mathbf{875 \pm 92}$ | $736 \pm 247$ |
| Cheetah, run | $438 \pm 33$ | $469 \pm 51$ | $\mathbf{495 \pm 13}$ | $454 \pm 41$ |
| Walker, walk | $414 \pm 310$ | $473 \pm 264$ | $\mathbf{597 \pm 102}$ | $533 \pm 98$ |
| Ball in cup, catch | $669 \pm 310$ | $910 \pm 58$ | $939 \pm 9$ | $\mathbf{944 \pm 22}$ |
| Mean | 613.7 | 717.3 | **767.3** | 720.5 |
| Median | 613.0 | 797.5 | **833.0** | 758.5 |

**Cube size.** Our space-time cube can be flexibly designed. We investigate the influence of temporal depth $k$ and the spatial size ($h$ and $w$, $h = w$ by default). The results based on 3 random seeds are shown in Figure 4. In general, a proper cube size leads to good results. The spatial size has a large influence on the final performance. A moderate spatial size (*e.g.*, $10 \times 10$) is good for MLR . The performance generally has an upward tendency when increasing the cube depth $k$. However, a cube mask with too large $k$ possibly masks some necessary contents for the reconstruction and hinders the training.

Table 5: Comparison of non-pretraining *Baseline* (*i.e.*, *Baseline*), MLR-pretrained *Baseline* (denoted as *MLR-Pretraining*) and our joint learning MLR (denoted as *MLR-Auxiliary*) on DMControl-100k.

| DMControl-100k | Cheetah, run | Reacher, easy |
|---|---|---|
| Baseline | $438 \pm 33$ | $557 \pm 137$ |
| MLR-Pretraining | $468 \pm 27$ | $862 \pm 180$ |
| MLR-Auxiliary | $\mathbf{495 \pm 13}$ | $\mathbf{875 \pm 92}$ |

Table 6: Cosine similarities of the learned representations from the masked observations and those from the original observations in *Baseline* and MLR.

| Cosine Similarity | Cheetah, run | Reacher, easy |
|---|---|---|
| Baseline | 0.366 | 0.254 |
| MLR | **0.930** | **0.868** |

## C.3 Discussion

**More analysis on MLR-S, MLR-T and MLR.** *MLR-S* masks spatial patches for each frame independently while *MLR-T* performs masking of entire frames. *MLR-S* and *MLR-T* enable a model to capture rich spatial and temporal contexts, respectively. We find that some tasks (*e.g.*, *Finger-spin* and *Walker-walk*) have more complicated spatial contents than other tasks on the DMControl benchmarks, requiring stronger spatial modeling capacity. Therefore, *MLR-S* performs better than *MLR-T* on these tasks. While for tasks like *Cartpole-swingup* and *Ball-in-cup-catch*, where objects have large motion dynamics, temporal contexts are more important, and *MLR-T* performs better. *MLR* using space-time masking harnesses both advantages in most cases. But it is slightly inferior to *MLR-S/-T* in *Finger-spin* and *Cartpole-swingup* respectively, due to the sacrifice of full use of spatial or temporal context as in *MLR-S* or *MLR-T*.

**Evaluation on learned representations.** We evaluate the learned representations from two aspects: **(i) Pretraining evaluation**. We conduct pretraining experiments to test the effectiveness of the learned representation. ***Data collection***: We use a 100k-steps pretrained *Baseline* model to collect 100k transitions. ***MLR pretraining***: We use the collected data to pretrain MLR without policy learning (*i.e.*, only with MLR objective). ***Representation testing***: We compare two models on DMControl-100k, *Baseline* with the encoder initialized by the MLR-pretrained encoder (denoted as *MLR-Pretraining*) and *Baseline* without pretraining (*i.e.*, *Baseline*). The results in Table 5 show that *MLR-Pretraining* outperforms *Baseline* which is without pretraining but still underperforms MLR which jointly learns the RL objective and the MLR auxiliary objective (also denoted as *MLR-Auxiliary*). This validates the importance of the learned state representations more directly, but is not the best practice to take the natural of RL into account for getting the most out of MLR. This is because that RL agents learn from interactions with environments, where the experienced states vary as the policy network is updated. **(ii) Regression accuracy test.** We compute the cosine similarities between the learned representations from the masked observations and those from the corresponding original observations (*i.e.*, observations without masking). The results in Table 6 show that there are high cosine similarity scores of the two representations in MLR while low scores in *Baseline*. This indicates that the learned representations of MLR are more predictive and informative.

**Performance on more challenging control tasks.** We further investigate the effectiveness of MLR on more challenging control tasks such as *Reacher-hard* and *Walker-run*. We show the test scores based on 3 random seeds at 100k and 500k steps in Table 7. Our MLR still significantly outperforms *Baseline*.

**The relationship between PlayVirtual and our MLR.** The two works have a consistent purpose, *i.e.*, improving RL sample efficiency, but address this from two different perspectives. PlayVirtual focuses on how to generate more trajectories for enhancing representation learning. In contrast, our MLR focuses on how to exploit the data more efficiently by promoting the model to be predictive of the spatial and temporal context through masking for learning good representations. They are compatible and have their own specialities, while our MLR outperforms PlayVirtual on average.

Table 7: Comparison of *Baseline* and MLR on more challenging DMControl tasks.

| Steps | Model | Reacher, hard | Walker, run |
|-------|-------|---------------|-------------|
| 100k | Baseline | $341 \pm 275$ | $105 \pm 47$ |
|  | MLR | $\mathbf{624 \pm 220}$ | $\mathbf{181 \pm 19}$ |
| 500k | Baseline | $669 \pm 290$ | $466 \pm 39$ |
|  | MLR | $\mathbf{844 \pm 129}$ | $\mathbf{576 \pm 25}$ |

**Application and limitation.** While we adopt the proposed MLR objective to two strong baseline algorithms (*i.e.*, SAC and Rainbow) in this work, MLR is a general approach for improving sample efficiency in vision-based RL and can be applied to most existing vision-based RL algorithms (*e.g.*, EfficientZero [7] [30]). We leave more applications of MLR in future work. We have shown the effectiveness of the proposed MLR on multiple continuous and discrete benchmarks. Nevertheless, there are still some limitations to MLR. When we take a closer look at the performance of MLR and *Baseline* on different kinds of Atari games, we find that MLR brings more significant performance improvement on games with backgrounds and viewpoints that do not change drastically (such as *Qbert* and *Frostbite*) than on games with drastically changing backgrounds/viewpoints or vigorously moving objects (such as *Crazy Climber* and *Freeway*). This may be because there are low correlations between adjacent regions in spatial and temporal dimensions on the games like *Crazy Climber* and *Freeway* so that it is more difficult to exploit the spatial and temporal contexts by our proposed mask-based latent reconstruction. Besides, MLR requires several hyperparameters (such as mask ratio and cube shape) that might need to be adjusted for particular applications.

## D   Broader Impact

Although the presented mask-based latent reconstruction (MLR) should be categorized as research in the field of RL, the concept of reconstructing the masked content in the latent space may inspire new approaches and investigations in not only the RL domain but also the fields of computer vision and natural language processing. MLR is simple yet effective and can be conveniently applied to real-world applications such as robotics and gaming AI. However, specific uses may have positive or negative effects (*i.e.*, the dual-use problem). We should follow the responsible AI policies and consider safety and ethical issues in the deployments.

---

[7]EfficientZero augments MuZero [19] with an auxiliary self-supervised learning objective similar to SPR [20] and achieves a strong sample efficiency performance on Atari games. We find that EfficientZero and our MLR have their skilled games on the Atari-100k benchmark. While EfficientZero wins more games than MLR (EfficientZero 18 versus MLR 8), the complexity of EfficientZero is much higher than MLR. Our MLR is a generic auxiliary objective and can be applied to EfficientZero. We leave the application in future work.

Table 8: Hyperparameters used for DMControl.

| Hyperparameter | Value |
|---|---|
| Frame stack | 3 |
| Observation rendering | (100, 100) |
| Observation downsampling | (84, 84) |
| Augmentation | Random crop and random intensity |
| Replay buffer size | 100000 |
| Initial exploration steps | 1000 |
| Action repeat | 2 *Finger-spin* and *Walker-walk*; |
| | 8 *Cartpole-swingup*; |
| | 4 otherwise |
| Evaluation episodes | 10 |
| Optimizer | Adam |
| $\quad (\beta_1, \beta_2) \rightarrow (\theta_f, \theta_\phi, \theta_g, \theta_q, \theta_\omega)$ | (0.9, 0.999) |
| $\quad (\beta_1, \beta_2) \rightarrow (\alpha)$ (temperature in SAC) | (0.5, 0.999) |
| Learning rate $(\theta_f, \theta_\omega)$ | 0.0002 *Cheetah-run* |
| | 0.001 otherwise |
| Learning rate $(\theta_f, \theta_\phi, \theta_g, \theta_q)$ | 0.0001 *Cheetah-run* |
| | 0.0005 otherwise |
| Learning rate warmup $(\theta_f, \theta_\phi, \theta_g, \theta_q)$ | 6000 steps |
| Learning rate $(\alpha)$ | 0.0001 |
| Batch size for policy learning | 512 |
| Batch size for auxiliary task | 128 |
| Q-function EMA $m$ | 0.99 |
| Critic target update freq | 2 |
| Discount factor | 0.99 |
| Initial temperature | 0.1 |
| Target network update period | 1 |
| Target network EMA $m$ | 0.9 *Walker-walk* |
| | 0.95 otherwise |
| State representation dimension $d$ | 50 |
| **MLR Specific Hyperparameters** | |
| Weight of MLR loss $\lambda$ | 1 |
| Mask ratio $\eta$ | 50% |
| Sequence length $K$ | 16 |
| Cube spatial size $h \times w$ | $10 \times 10$ |
| Cube depth $k$ | 4 *Cartpole-swingup* and *Reacher-easy* |
| | 8 otherwise |
| Decoder depth $L$ (number of attention layers) | 2 |

Table 9: Hyperparameters used for Atari.

| Hyperparameter | Value |
| --- | --- |
| Gray-scaling | True |
| Frame stack | 4 |
| Observation downsampling | (84, 84) |
| Augmentation | Random crop and random intensity |
| Action repeat | 4 |
| Training steps | 100K |
| Max frames per episode | 108K |
| Reply buffer size | 100K |
| Minimum replay size for sampling | 2000 |
| Mini-batch size | 32 |
| Optimizer | Adam |
| Optimizer: learning rate | 0.0001 |
| Optimizer: $\beta_1$ | 0.9 |
| Optimizer: $\beta_2$ | 0.999 |
| Optimizer: $\epsilon$ | 0.00015 |
| Max gradient norm | 10 |
| Update | Distributional Q |
| Dueling | True |
| Support of Q-distribution | 51 bins |
| Discount factor | 0.99 |
| Reward clipping Frame stack | [-1, 1] |
| Priority exponent | 0.5 |
| Priority correction | $0.4 \rightarrow 1$ |
| Exploration | Noisy nets |
| Noisy nets parameter | 0.5 |
| Evaluation trajectories | 100 |
| Replay period every | 1 step |
| Updates per step | 2 |
| Multi-step return length | 10 |
| Q network: channels | 32, 64, 64 |
| Q network: filter size | $8 \times 8, 4 \times 4, 3 \times 3$ |
| Q network: stride | 4, 2, 1 |
| Q network: hidden units | 256 |
| Target network update period | 1 |
| $\tau$ (EMA coefficient) | 0 |
| **MLR Specific Hyperparameters** | |
| Weight of MLR loss $\lambda$ | 5 *Pong* and *Up N Down* |
| | 1 otherwise |
| Mask ratio $\eta$ | 50% |
| Sequence length $K$ | 16 |
| Cube spatial size $h \times w$ | $12 \times 12$ |
| Cube depth $k$ | 8 |
| Decoder depth $L$ (number of attention layers) | 2 |