# OpenReview forum: "Mask-based Latent Reconstruction for Reinforcement Learning"
_NeurIPS.cc/2022/Conference — NeurIPS 2022 Accept_

### Official Review · Reviewer_du6d · 2022-06-29

**Rating:** 5
**Confidence:** 4
**Soundness:** 3 good
**Presentation:** 3 good
**Contribution:** 2 fair

**Summary:**

The paper tackles the field of state representation learning in visual RL. They propose Mask-based Latent Reconstruction (MLR) for learning state representations based on masking sequences of input images. This practice is inspired by the computer vision and natural language processing communities. The approach is different than the ones used traditionally in these fields by reconstructing the missing content in the latent space instead of the original one. The authors provide an extensive evaluation to demonstrate the improvement in performance their method achieves to SOTA methods. Their evaluation includes discrete and continuous control environments as well as ablations of the different design decisions introduced in their approach.

**Questions:**

- If I understood correctly, "baseline" in your tables refers to either SAC or rainbow. In table 1, that should be SAC. Can you explain how SAC is outperforming other methods which add on top of it such as SAC-AE and CURL? these methods were built with the motivation that SAC on pixels isn't good enough. Can you explain how these results relate to the original ones from those papers?
- this brings me to the next question: how do you ensure a fair comparison in your evaluation? do all the methods use the same SAC implementation, same encoder, decoder architectures, and hyperparameters, etc...? Please elaborate on the standardization of experiments.
- Related to what I mentioned in the weakness section, can you elaborate on the intuition behind your choice of targets?  to me, this seems like a random choice. In other words, what are you trying to enforce on the state representation? what kind of property? and how does your method allow for this property to emerge?  For instance, when using autoencoders, the property is compressing, with contrastive methods, it's somehow clustering... Please give a clear explanation here and not a high-level explanation.
- Related to the previous comment, if MRL is not integrated with policy learning and used instead for pretaining only, would it actually help? I know that one of the motivations of the approach is that the two objectives are simultaneously and jointly trained, but such a test would be important to validate the importance of the learned state representation. Can you include such an experiment in the final version? for the rebuttal it doesn't have to be over all the environments...(whatever can be managed).

**Limitations:**

The paper lacks an evaluation of the obtained state representations. An evaluation could be something like the regression accuracy (or a correlation of the two variables) of the original states (provided by simulation) based on the representation. The pertaining experiment I mentioned in my last question would also qualify as a (qualitative) evaluation of the obtained representations.

**Strengths And Weaknesses:**

Strengths:
========

- The idea of using masked observations for state representation learning is to my knowledge novel in RL
- The paper is well situated with respect to previous related work
- The paper is mostly (with few exceptions) well-written and easy to follow (clear)
- The proposed method is simple to implement
- The evaluation is extensive: it includes multiple baselines, different kinds of environments (discrete and continuous action spaces), and ablations of the different components of the method
- The results show that the proposed method mostly outperforms the baselines and also leads to a low reward variance.

Weaknesses:
==========

- In section 3.2, paragraph iii) (seconding) should be rewritten and made clearer. Multiple details in this paragraph are ambiguous and hard to follow. The first sentence in this paragraph is itself very confusing and could be interpreted in many ways (what is meant by "visible ones" and "implicit way"). The motivation for using the actions is missing or at least not convincing. Understanding relative positional embedding requires reading [43], which makes the paper not self-contained. Additionally, it's unclear what the plus in equation 2 means, is it an actual addition or a concatenation? it can't really be an addition since the two vectors have different dimensions, so is it a concatenation or something else? please specify.
- The proposed approach claims to reconstruct the masked content in the latent space. This is done using the cosine loss based on a target that is obtained using the original images and a moving average of the online encoder and projection head. It is unclear to me what is the meaning of such a loss, why would we want to push the representation to be similar to another one produced by a moving average over its previous weights but using the original images? I would have been more convinced if the method simply relied on reconstructing the original pixels as done in the ablation. Let me rephrase, I'm not saying that the method doesn't make sense, I'm just saying that the intuition behind the state representation learning method is unclear (at least to me). I'm willing to change my rating if this aspect is made clearer and of course convincing.

Minor Problems:
============

- the paper repeatedly mentions that the presented approach reconstructs the masked content in the latent space, which is a false statement. The method attempts to reconstruct the information available in the masked content in the latent space.
- some typos and grammar problems here and there. I would recommend a non-author to go over the paper + some grammar check software. e.g. in line 262 neural --> natural.

---

> ### Author Response · Authors · 2022-08-02
> **Response to Reviewer du6d (Part 1)**
>
> Thank you for the positive comments on our idea, experiment and presentation. We address your concerns below. The contents we revised in our main submission and appendix are marked in red.
>
> **Q1: Revision of Section 3.2 (iii) Decoding.**
>
> A1: Thanks for your suggestion. We have rewritten Section 3.2 (iii) as suggested for clarity. **(1) About the confusing statements:** The "visible ones" refers to unmasked contents and we have revised it. The "implicit way" means that we reconstruct the information corresponding to the masked contents in the latent space instead of reconstructing pixels explicitly. We have discarded this word in our updated description to avoid confusion. **(2) Motivation of using the actions:** In RL, the next state is determined by both the current state and the action at this time step. By considering this nature of RL, we utilize the actions as assistive contexts to reduce the ambiguity in prediction. Table 3 shows the ablation study on the use of action tokens. Thanks to the reduction of ambiguity, the results show that using action tokens from MLR brings in obvious performance improvement. **(3) About positional embedding:** The positional information of each token is firstly encoded as a vector that has the same dimension with the original token. Then, this vector is added into its corresponding token via an element-wise summation operation, i.e., the plus in equation 2.
>
> **Q2: "It is unclear to me what is the meaning of such a loss, why would we want to push the representation to be similar to another one produced by a moving average over its previous weights but using the original images?"**
>
> A2: We decouple your this question into two sub-ones to address as follows. **(1) Why do we reconstruct the latent representations instead of reconstructing original images?** Reconstructing images is a basic solution but not a well-designed one for RL problems as indicated by our experimental results in Table 2. This is because the original images are signals of high information density as analyzed in [1], which may contain distractions and redundancies for the policy learning in RL. Besides, reconstructing images is less directly to supervise the learned latent representation used for policy learning than reconstructing the latent representations. **(2) Why do we adopt a moving averaging model to produce the reconstruction targets?** For the two latent representations to be pushed to be similar, one is the reconstruction result while the other is its corresponding target. Adopting a moving averaging model to produce reconstruction targets is a widely used technique in self-supervised learning (e.g., BYOL [2]) where the moving averaging model serves as a means of stabilizing the bootstrap step to provide more stable prediction targets.
>
> [1] He, Kaiming, et al. "Masked autoencoders are scalable vision learners." Proceedings of the IEEE/CVF Conference on Computer Vision and Pattern Recognition. 2022.
>
> [2] Grill, Jean-Bastien, et al. "Bootstrap your own latent-a new approach to self-supervised learning." Advances in neural information processing systems 33 (2020): 21271-21284.
>
> **(Minor) Q3: "The paper repeatedly mentions that the presented approach reconstructs the masked content in the latent space, which is a false statement. The method attempts to reconstruct the information available in the masked content in the latent space."**
>
> A3: Thanks for pointing this out. We have modified the statement in the revision as you suggested.
>
> **(Minor) Q4: Typos.**
>
> A4: Thank you for the suggestions. We have fixed the typos.
>
> **Q5: Why can our Baseline outperform SAC+AE and CURL?**
>
> A5: Compared to naive SAC or Rainbow on pixels, our Baseline additionally adopts data augmentation (random crop and random intensity) on top of these baselines to enable them to outperform other methods you mentioned. This practice follows prior works [1, 2] which uncovers that applying proper data augmentation can significantly improve the sample efficiency of SAC so that it is superior to other methods such as SAC-AE and CURL.
>
> [1] Laskin, Misha, et al. "Reinforcement learning with augmented data." NeurIPS 2020.
>
> [2] Yarats, Denis, et al. "Image augmentation is all you need: Regularizing deep reinforcement learning from pixels." ICLR 2020.
>
> **Q6: How do you ensure a fair comparison in your evaluation? do all the methods use the same SAC implementation, same encoder, decoder architectures, and hyperparameters, etc...?**
>
> A6: Yes, for the compared SOTA methods with the same baselines (i.e., SAC and rainbow), such as SAC+AE, CURL, DrQ, PlayVirtual, we use the same configurations including implementation, architectures, and hyperparameters with those of our MLR for fair comparison. Besides, we also directly compare our MLR with its corresponding Baselines to demonstrate the effectiveness of our introduced auxiliary objective, i.e., MLR.

---

> > ### Author Response · Authors · 2022-08-02
> > **Response to Reviewer du6d (Part 2)**
> >
> > **Q7: Clarification of the choice of reconstruction targets.**
> >
> > A7: Our choice of reconstruction targets is not a random choice. The reconstruction results and targets are in one-to-one correspondence according to the timestamps of observation sequence. From an masked observation in a sequence, we predict the state representations of its corresponding original observation (i.e., the observation without any contents masked) in the same sequence where **the state representations of the original observations inferred by a moving averaging model are deterministically chosen as the reconstruction targets**. The correlations of different contents along the spatial and temporal dimensions enforce the predictability of the state representations and the consistency between the state representations of the masked observations and those of the original observations. We are not sure whether we fully understand your question about the property. If we have not, welcome your further comments and discussion.
> >
> > **Q8: Validating the importance of the learned state representations.**
> >
> > A8: We appreciate your insightful suggestion on validating the importance of the learned state representations by utilizing our MLR as a pretraining objective. Partial results on DMControl-100K available during the rebuttal are shown below and added in our revised appendix. The results over more environments will be updated into our final version to provide more in-depth insights. The configurations of the pretraining and joint-training models are listed below:
> > *MLR-Pretraining:* We firstly train a *Baseline* model (without MLR) to collect 100k transitions, then adopt these collected transitions to pre-train the RL agent with MLR objective where no policy learning is integrated. Afterwards, initializing the RL agent with the MLR-pretrained weights, we train the agent for the downstream policy learning.
> > *MLR-Auxiliary (i.e., the MLR in our paper):* We treat MLR as an auxiliary objective for the policy learning of RL in a joint training pipeline as introduced in our main submission.
> >
> > | DMControl-100k  | Cheetah, run | Reacher, easy
> > | :-----| :----:   | :----:   |
> > | Baseline (without MLR) | 438 ± 33 | 557 ± 137 |
> > | MLR-Pretraining | 468 ± 27 | 862 ± 180 |
> > | MLR-Auxiliary | **495 ± 13** | **875 ± 92** |
> >
> > As shown above, **treating our MLR as a pretraining objective is also helpful when compared to the baseline model.** It indeed validates the importance of the learned state representations for the policy learning more directly, but is not the best practice to take the nature of RL into account for making the most of our proposed MLR. This is because RL agents learn from interactions with environments, where the experienced states vary as the policy network is updated. Thus, collecting additional transitions for pretraining is not only costly but also of high difficulty to collect transitions whose corresponding state space well matches with the state space in the downstream policy learning.
> >
> > **Q9: Evaluating the quality (regression accuracy) of the learned state representations.**
> >
> > A9: We evaluate the quality of the learned state representations by calculating the cosine similarity between the learned representations from the masked observations and those from the corresponding original observations (i.e., observations without masking). The results in the table below show that there is a high correlation of the two representations in MLR while a low correlation in Baseline. This indicates that learned representations by MLR are more predictive and informative.
> > | Cosine Similarity  | Cheetah, run | Reacher, easy
> > | :-----| :----:   | :----:   |
> > | Baseline | 0.366 | 0.254  |
> > | MLR | **0.930** | **0.868** |

---

> > > ### Comment · Reviewer_du6d · 2022-08-05
> > > **Response to authors**
> > >
> > > Thank you for the response and the additional evaluations. Your answers addressed many of my concerns. A couple more things:
> > >
> > > - **Concerning (Q5)**, this should be mentioned when introducing the baseline.
> > >
> > > - **Concerning (Q9)**, this test is interesting in comparison to other baselines including state representation learning methods (e.g. CURL, dreamer...), can you provide the metrics for these methods?

---

> > > > ### Author Response · Authors · 2022-08-06
> > > > **Response to Reviewer du6d**
> > > >
> > > > Many thanks for your further comments and suggestions.
> > > >
> > > > **Reply to Concerning (Q5)**. We have clarified this as you suggest in our revised main paper when introducing our Baseline. Please see the Section 4.1: “Implementation” (marked in red) for more details.
> > > >
> > > > **Reply to Concerning (Q9)**. Thank you for this suggestion. We further compare our MLR with two representative methods, i.e., CURL and PlayVirtual, in the qualities of their learned state representations. The CURL enhanaces the learned representations spatially by improving the instance discrimination via contrastive learning while the PlayVirtual enhances the learned representations temporally by dynamics prediction. As shown in the experiment results below, our MLR has the highest regression accuracy, which suggests our MLR is also superior to these methods in enhancing the learned state representations by the proposed spatiotemporal mask-based latent reconstruction. We will try our best to report more comparison results in our final version due to the time limitation during this rebuttal.
> > > >
> > > > | Cosine Similarity  | Cheetah, run | Reacher, easy
> > > > | :-----| :----:   | :----:   |
> > > > | Baseline | 0.366 | 0.254  |
> > > > | CURL | 0.529 | 0.431  |
> > > > | PlayVitual | 0.707 | 0.792 |
> > > > | MLR | **0.930** | **0.868** |

---

> > > > > ### Author Response · Authors · 2022-08-08
> > > > > **Looking forward to your further feedbacks**
> > > > >
> > > > > Many thanks for your efforts in reviewing our work. We are trying our best to address your concerns. We would appreciate it if you could let us know whether you have other concerns so that we could have the opportunity to address them before the end of this rebuttal.

---

> > > > > > ### Comment · Reviewer_du6d · 2022-08-08
> > > > > > **No further concerns**
> > > > > >
> > > > > > Thank you for being responsive to the feedback and for addressing my concerns. I now raised my score. Please note that the paper still has some typos and grammar errors (e.g. sentences starting with 'And'). Please make sure that the final version is well-polished.

---

> > > > > > > ### Author Response · Authors · 2022-08-08
> > > > > > > **Reply to Reviewer du6d**
> > > > > > >
> > > > > > > Thank you very much for your responsible efforts in reviewing! We will make sure the final version is carefully polished.

---

### Official Review · Reviewer_djAT · 2022-07-10

**Rating:** 7
**Confidence:** 4
**Soundness:** 3 good
**Presentation:** 4 excellent
**Contribution:** 2 fair

**Summary:**

The paper introduces mask based reconstruction (MLR) for RL with image inputs. MLR is an auxiliary objective optimized together with the policy learning objective. The idea is to reconstruct masked pixels in the latent state instead of the input space where state representations are inferred from original unmasked frames. The paper proposes to mask portions on both spatial and temporal dimensions in the observation sequence. There is an “online encoder” for masked observations and a momentum-encoder for original images. Furthermore, there is a Transformer-based latent decoder that also takes action embeddings as input. MLR uses a projection + prediction head to get the final prediction result, and cosine similarity to compute the distance between masked and original sequence. The paper presents several ablations about the masking strategies, depth of decoder, similarity losses, etc. Results on DMC-100K and Atari-100K show that MLR outperforms leading algorithms like SPR, PlayVirtual.


**Questions:**

1. The method was not compared with EfficientZero which is an important baseline. How does MLR compare with EfficientZero?
2. The Appendix talks about the training time and details. How they compare with leading methods like SPR?
3. There are no discussions on scenarios where the mask-based reconstruction is helping. For example, are improvements on games like MsPacman where the background is not moving or games with moving viewpoints like Crazy Climber? This could also help understand the limitations of the proposed method.

**Ethics Review Area:**

["I don’t know"]

**Limitations:**

The paper does not highlight the limitations of MLR. The paper should talk about potential bottlenecks when scaling MLR to more complex tasks.

**Strengths And Weaknesses:**

### Strengths
- Paper is well written and easy to follow.
- Results and ablation studies are thorough and demonstrates the efficacy of each component.
- Interesting direction to apply mask-based reconstructions in latent space as it is challenging to learn good latent representations while solving RL tasks.

### Weaknesses
- Novelty is limited as Masked based reconstruction are already known to work well in Vision tasks.
- Limitations of the proposed method are not discussed.

---

> ### Author Response · Authors · 2022-08-02
> **Response to Reviewer djAT**
>
> Thanks for your valuable suggestions and positive comments on our idea, results, ablations and writing. We address your remaining concerns below. The contents we revised in our main submission and appendix are marked in red.
>
> **Q1: Novelty and contribution of MLR.**
>
> A1: Mask-based reconstruction is known to work well in vision tasks. However, how to efficiently design mask-based reconstruction for RL is still under-explored. To the best our knowledge, we are the first to explore mask-based video modeling in the RL community and make a systematical empirical study to investigate the good practices of masking and reconstruction in RL. First, previous mask-based reconstruction in vision tasks focus on image. For RL, the consecutive frames are highly correlated and we investigate spatiotemporal mask-based reconstruction for observation sequences (video clips). Second, previous works reconstruct missing content in the original space (e.g., RGB pixels). We found that for RL task which leverages the latent feature for policy learning, reconstructing the missing information in the latent space is more efficient.
>
> **Q2: Discussion on the limitations of MLR.**
>
> A2: Thanks for pointing this out. The limitations of MLR mainly lie in (i) not excelling at controlling games with drastically changing backgrounds/viewpoints or objects of high motion dynamics such as Crazy Climber and Freeway where there are low correlations between adjacent regions in spatial and temporal dimensions; and (ii) requiring several hyperparameters to adjust for specific applications. We have added more discussion on the limitations in Appendix C.3: "Application and limitation" in the revision.
>
> **Q3: Comparison with EfficientZero.**
>
> A3: Sorry for not comparing Rainbow-based MLR with MuZero-based EfficientZero on Atari games given that they use inconsistent baselines and fall into different branches of RL methods (i.e., model-free and model-based RL, respectively). We found that MLR and EfficientZero have their skilled games on the Atari-100k benchmark. While EfficientZero wins more games than MLR (EfficientZero 18 v.s. MLR 8), the complexity of EfficientZero is much higher than MLR. Besides, MLR is a generic auxiliary objective and can be applied to EfficientZero. We will consider to add our MLR objective to EfficientZero in future revisions.
>
> **Q4: Training time details.**
>
> A4: Thank you for pointing this out. We tested the wall-clock training time with Nvidia GeForce RTX 3090 GPU. On Atari, the training time is 6.0, 10.9, 4.0, and 8.2 hours for SPR, PlayVirtual, Baseline and MLR, respectively. On DMControl, the training time is 4.3, 5.2, 3.8, 6.5 hours for SPR, PlayVirtual, Baseline and MLR, respectively. We will leave the optimization of our training strategy as future work to speed up the training, e.g., reducing the frequency of using masked-latent reconstruction. We have added the details in Appendix B.2: "GPU and wall-clock time" in the revision.
>
> **Q5: Discussion on scenarios where MLR is helping and not helping.**
>
> A5: Thank you for the insightful comments and suggestions. MLR brings more significant performance improvement on games with backgrounds and viewpoints that do not change drastically (such as Qbert and Frostbite) than on the games with drastically changing backgrounds/viewpoints or vigorously moving objects (such as Crazy Climber and Freeway). This may be because there are low correlations between adjacent regions in spatial and temporal dimensions on the games like Crazy Climber and Freeway so that it is more difficult to exploit the spatial and temporal contexts by our proposed mask-based latent reconstruction.

---

> > ### Comment · Reviewer_djAT · 2022-08-08
> > **Post Rebuttal Comments**
> >
> > I appreciate the authors for answering to my questions and I am increasing the score by 1. Some comments based on the response are:
> >
> > - I hope the authors add the limitations in the revision. Regarding the first limitation- Does this mean that the proposed method might not do well when the the agent viewpoint is changing with its action (like in scenarios where the agent only observes the part of the arena)?
> > - I feel adding MLR objective to EfficientZero should be discussed in future works section along with the comparison mentioned in above.
> > - Regarding discussion of scenarios where MLR helps or not: I hope the authors discuss this in the paper in detail.

---

> > > ### Author Response · Authors · 2022-08-08
> > > **Response to Reviewer djAT**
> > >
> > > We appreciate your positive comments and very valuable suggestions on our work! Our proposed method is generally applicable. It can still handle but does not gain as that much in the environments where background/viewpoint changes are very drastic (the spatial prediction capability still helps while the temporal prediction capability is less helpful). We will add this disscussion and further carefully improve our paper in its final version following your detailed suggestions.

---

### Official Review · Reviewer_ZSQ9 · 2022-07-10

**Rating:** 6
**Confidence:** 4
**Soundness:** 2 fair
**Presentation:** 3 good
**Contribution:** 3 good

**Summary:**

This paper proposes a mask-based latent reconstruction/prediction auxiliary task for visual RL. The proposed auxiliary task takes as input a sequence of observations, applies a uniformly sampled spatiotemporal mask to the sequence, and then task the network with predicting the latent representation of the unmasked sequence encoded by an EMA of the online encoder. The proposed method, MLR, is evaluated on DMControl and Atari tasks, and a number of ablation studies uncover the key design choices of the proposed framework.

**Questions:**

I believe I have already made my biggest concerns clear in the "weaknesses" section above. However, I do have a few questions for the authors that should be addressable during the rebuttal:
- It is not clear to me after reading the paper + appendix exactly which data augmentation is used and where. Can you elaborate on this?
- I'd also like the authors to clarify on how the "Baseline" SAC implementation differs from that of other published works. Appendix describes the architecture to be based on CURL; would "Baseline" be equivalent to CURL w/o the auxiliary loss? Is data augmentation applied?
- How do the listed wall-times for the proposed method compare to (1) the "Baseline" implementation without the auxiliary loss, and (2) prior work such as CURL, DrQ, SPR, or PlayVirtual?

**Limitations:**

I believe the authors provide a satisfactory discussion of potential negative societal impact of their work, but discussion of limitations is minimal. I'd like to see more explicit discussion of when the proposed auxiliary loss can be expected to be useful, and when it may fail (e.g., which key assumptions are made).

**Strengths And Weaknesses:**

Strengths:
- Conceptually simple, clearly presented, and ablations are insightful.
- Experimental results are encouraging, the provided rliable metrics are appreciated, and code is provided.

Weaknesses:
- DMControl evaluation. I understand that the particular choice of DMControl tasks used in the evaluation is convenient due to the number of available baseline results. However, performance on these tasks have largely been saturated by prior work and are thus not a good indicator of algorithmic improvements (e.g., does a median performance of 914 vs 929.5 vs 935 vs 907 vs 948 @ 500k steps really matter? I understand that 100k steps may be more informative than 500k but point still stands, and the improvement of curves for these tasks in Figure 2 of appendix don't appear significant to me, presumably for the same reason). I strongly suggest the authors evaluate on more challenging tasks / benchmarks in future revisions, although I realize that this may not be feasible within the strict time frame of a rebuttal.
- Related work. I would like to see more discussion of RL methods that use BYOL-style auxiliary objectives and data augmentation/masking in the related work section, given the strong similarity to such methods. I will be happy to provide references, but am under the impression that the authors are familiar with the literature given some of these works (e.g. SPR) are cited and/or briefly mentioned in the experiment section.

**Post-rebuttal:** In light of the author response I am willing to raise my score by 1. I expect the authors to include reviewer suggestions in a future revision.

---

> ### Author Response · Authors · 2022-08-02
> **Response to Reviewer ZSQ9**
>
> Thank you for the valuable suggestions and positive comments on this work as “conceptually simple, clearly presented, and ablations are insightful”. We address your remaining concerns below. The contents we revised in our main submission and appendix are marked in red.
>
> **Q1: About the DMControl evaluation.**
>
> A1: Thanks for your helpful suggestions. Yes, more challenging tasks should be used in the community for evaluation. We conduct experiments on two additional tasks that are more difficult than the six tasks we already used in the standard sample-efficiency DMControl benchmarks. The results in the table below show more significant performance gains brought by our proposed method. We have add the experiments in Appendix C.3 in the revision. We will report the results on more challenging tasks to show the improvements of our proposed MLR in the future revision.
>
> | Steps  | Model| Reacher, hard | Walker, run
> | :-----| :----:   | :----:   | :----:   |
> | 100k| Baseline  | 341 ± 275 | 105 ± 47  |
> | 100k | MLR | **624 ± 220** | **181 ± 19** |
> | 500k| Baseline | 669 ± 290 | 466 ± 39  |
> | 500k | MLR | **844 ± 129** | **576 ± 25** |
>
>
> **Q2: More discussion in related work.**
>
> A2: Thanks for your suggestions. Inspired by the success of BYOL in learning visual representations, recent works (e.g., SPR, PlayVirtual, Proto-RL, SODA and BYOL-Explore) introduce BYOL-style learning objectives to vision-based RL for learning effective state representations and show promising performance. The BYOL-style learning is often integrated into auxiliary objectives in RL such as future state prediction as in SPR and BYOL-Explore, cycle-consistent dynamics prediction as in PlayVirtual, prototypical representation learning as in Proto-RL and invariant representation learning as in SODA. These BYOL-style auxiliary objectives are commonly trained with data augmentation since it can conveniently produce two BYOL views for the bootstrap learning scheme. For example, SPR and PlayVirtual apply random crop and random intensity to input observations in Atari games. The proposed MLR auxiliary objective can be categorized into BYOL-style auxiliary objectives. We have added the discussions on the related works including BYOL, BYOL-style auxiliary objectives in RL and data augmentation in our revised main submission and appendix.
>
>
>
> **Q3: Elaboration on data augmentation.**
>
> A3: The data augmentation strategies adopted for DMControl and Atari have been listed as one of hyperparameter configurations in Table 9 and 10 of Appendix, respectively. In brief, we follow the common practices in this field to adopt random crop and random intensity on these two benchmarks.
>
> **Q4: Elaboration on Baseline implementation.**
>
> A4: Sorry for this confusion. Yes, our "Baseline" is equivalent to "CURL w/o the auxiliary loss" except for the slight differences on the applied data augmentation strategies. CURL adopts random crop for data augmentation. We follow the prior work SPR to adopt random crop and random intensity for both our Baseline and MLR. We have further clarified this implementation detail in Appendix B.2: "Baseline and data augmentation" in the revision.
>
> **Q5: Wall-clock training time comparison.**
>
> A5: Thanks for the suggestion. We tested the wall-clock training time with Nvidia GeForce RTX 3090 GPU. On Atari, the training time is 6.0, 10.9, 4.0, and 8.2 hours for SPR, PlayVirtual, Baseline and MLR, respectively. On DMControl, the training time is 4.3, 5.2, 3.8, 6.5 hours for SPR, PlayVirtual, Baseline and MLR, respectively. We will leave the optimization of our training strategy as future work to speed up the training, e.g., reducing the frequency of using masked-latent reconstruction.
>
> **Q6: More discussion on the limitations of MLR.**
>
> A6: Thank you for this suggestion. MLR brings more significant performance improvement on games with backgrounds and viewpoints that do not change drastically (such as Qbert and Frostbite) than on the games with drastically changing backgrounds/viewpoints or vigorously moving objects (such as Crazy Climber and Freeway). This may be because there are low correlations between adjacent regions in spatial and temporal dimensions on the games like Crazy Climber and Freeway so that it is more difficult to exploit the spatial and temporal contexts by our proposed mask-based latent reconstruction. We have extended the discussion on the limitations of MLR in Appendix C.3: "Application and limitation" in our revision.

---

> > ### Comment · Reviewer_ZSQ9 · 2022-08-04
> > **Response to authors**
> >
> > I appreciate your clarifications and promise of experiments on harder tasks. Wrt wall-time: I understand that numbers are implementation and hparam dependent, but it can be very helpful for readers to include this info since RL methods vary a lot in computational complexity. The author response largely addresses my concerns (aside from the DMControl experiments) and I am willing to raise my score with the expectation that the authors include these additional clarifications, discussion of related work, and experiments in a future revision.

---

> > > ### Author Response · Authors · 2022-08-05
> > > **Response to Reviewer ZSQ9**
> > >
> > > Thanks a lot for raising your score. We appreciate your responsible efforts in reviewing our work! We will include what contents you suggest in our final version to enable it to be a stronger paper.

---

### Official Review · Reviewer_uZPQ · 2022-07-10

**Rating:** 5
**Confidence:** 4
**Soundness:** 2 fair
**Presentation:** 3 good
**Contribution:** 2 fair

**Summary:**

This paper built upon recent advancement in NLP and CV to introduce an auxiliary task in RL. More specifically, the authors proposed to reconstruct latents from masked observations, with the goal of improving representations used in RL. A few experiments across both discrete and continuous control to demonstrate the promise of the proposed methods.

**Questions:**

1. The authors stated in L122 that MLR ``...is the theoretically applicable to different RL algorithms''. Is there any theory behind the proposed algorithm for the application in RL?

2. Do the features used for RL come from the output of online encoder, i.e., \tilde{s}?

**Limitations:**

Yes

**Strengths And Weaknesses:**

Strength:
1. Using the idea of auxiliary tasks to boost the RL performance has been explored and this paper belongs to this line of research. The idea from the proposed method looks reasonable and interesting.

2. The authors designed different types of experiments to justify the promise of the proposed method, and the results generally look promising.

3. This paper is mostly well written and organized.


Weakness:
1. One issue is that the current version lacks the explanation and investigation of why the proposed method should outperform the previous algorithms combining representation learning with RL, e.g., CURL. We can see some promise based on experimental results, but in-depth exploration on why MLR works is integral to make the claims convincing.

2. The numbers and plots in the experiments section definitely show that MLR achieves competitive performance. From Table 1, however, it seems that MLR has very similar performance when compared against PlayVirtual and CURL with 500k steps, which may suggest that the representations from MLR may not be sufficiently distinguishable from previous work.

3. My another concern regarding the experiments corresponds to Atari games. In Table 1 of the appendix, the authors showed that MLR can outperform Baseline, which denotes Rainbow. The issue is that the scores here are much lower when compared with those in the Rainbow paper. Just pick a simple one: For the game Pong, the reported score here is -3.1 while it's 19 in the Rainbow paper. Consequently, it's not convincing to me that the auxiliary task can help the RL task on Atari games.

4. A minor point regarding the background in Section 3.1: POMDP distinguishes state and observation in its formulation, but the text there lacks the discussion on states.

---

> ### Author Response · Authors · 2022-08-02
> **Response to Reviewer uZPQ**
>
> Thank you for the positive comments on our idea, experiment and writing. We address your concerns below. The contents we revised in our main submission and appendix are marked in red.
>
> **Q1: Explanation and investigation of why our proposed method can outperform previous works.**
>
> A1: The masked modeling methods have been widely investigated and been extensively demonstrated effective for enhancing the representation learning in the fields of Neural Language Processing (NLP) and Computer Vision (CV). Our proposed MLR makes the first endeavour to introduce a self-supervised masked modeling method for vision-based RL, which is still under-explored in this field. In vision-based RL, the consecutive video frames are highly correlated. Our proposed MLR outperforms previous works (e.g., CURL) because MLR effectively exploits such correlations along the spatial and temporal dimensions in learning informative representations by the auxiliary task design, i.e., predicting the masked information from the unmasked content of a video clip (observation sequence), while previous works (e.g., CURL) have not. In this way, MLR enforces the learned representations to be spatially and temporally predictive and consistent.
>
> **Q2: "The numbers and plots in the experiments section definitely show that MLR achieves competitive performance. From Table 1, however, it seems that MLR has very similar performance when compared against PlayVirtual and CURL with 500k steps."**
>
> A2: We aim at improving the sample efficiency of RL in this work. For this aspect, in Table 1, the results for 100k steps are more informative than those of 500k since the 100k results (where the amount of interaction data is not large for the task) reflect the sample efficiency of different methods while 500k results (more interaction data) tend to reflect the performance close to convergence. These evaluated tasks maybe not very challenging where 500k step interaction data already does not belong to low data regime. We conduct experiments on additional tasks that are more challenging. The results in the table below show the consistent effectiveness of our method even for 500k steps. We have add the experiments in Appendix C.3 in the revision. We will add more challenging tasks in the future revision. We also look forward to developing new benchmarks for sample efficiency evaluation in the future for the RL community.
>
> | Steps  | Model| Reacher, hard | Walker, run
> | :-----| :----:   | :----:   | :----:   |
> | 100k| Baseline  | 341 ± 275 | 105 ± 47  |
> | 100k | MLR | **624 ± 220** | **181 ± 19** |
> | 500k| Baseline | 669 ± 290 | 466 ± 39  |
> | 500k | MLR | **844 ± 129** | **576 ± 25** |
>
> **Q3: Clarification of Baseline's scores on Atari.**
>
> A3: The reported scores are different because they correspond to **different interaction steps**. The original Rainbow paper reports the results on 50M steps. Following the standard settings for evaluating the sample efficiency (i.e., Atari-100k benchmark), we report the scores on 100K steps (far less than 50M) in Table 1 of the appendix. The only difference of our MLR compared to Baseline lies in that we apply our proposed mask-based latent reconstruction on top of Baseline, which clearly shows the effectiveness of our design.
>
> **Q4: Discussion on state and observation in POMDP.**
>
> A4: Thanks for your suggestions. Following common practice [1], several consecutive image observations are stacked to construct a state to convert the POMDP to an MDP. We have updated the detailed description on the state and observation in Section 3.1. Please kindly see our revision for more details.
>
> [1] Mnih, Volodymyr, et al. "Playing atari with deep reinforcement learning." arXiv:1312.5602 (2013).
>
> **Q5: L122 that MLR ''...is the theoretically applicable to different RL algorithms''.**
>
> A5: To be more precise, we will replace "theoretically" with "generally" as "Our proposed MLR serves an auxiliary task applied on top of the regular RL training, which is generally applicable to different RL algorithms."
>
> **Q6: Do the features used for RL come from the output of online encoder?**
>
> A6: Yes, they are from the online encoder as in the prior works (CURL, SPR, PlayVirtual, etc.).

---

> > ### Author Response · Authors · 2022-08-08
> > **Looking forward to your feedbacks**
> >
> > Many thanks for your efforts in reviewing our work! We are making our best efforts to address your concerns and looking forward to your feedbacks on our responses. Please feel free to let us know whether our responses address your concerns and whether you have other questions or suggestions.

---

### Meta-Review · Area_Chair_taYf · 2022-08-29

**Recommendation:** Accept
**Confidence:** Less certain

**Metareview:**

Unanimous accept (scoring 5675, with confidence, from reviewers who have published in similar areas)

All four reviewers all agree that the work is novel (using such masked-based representatives/auxiliary losses in an RL setting in the _latent_ space opposed to the original RGB space) which doesn't sound like an obvious idea a-priori, but (all reviewers agree) the authors back this up with extensive experiments and ablations of their own method in Atari and DMC. The authors were very responsive with reviewers, resulting in 3 reviewers increasing their scores each by one. All reviewers agree this paper is easy to follow, and most of their stated weaknesses/confusions have now been addressed or clarified.

My view of this work is that the main contribution is a latent reconstruction loss, as an additional objective to whatever the RL task objective is. And this is useful for using data augmentation / self-supervised learning in RL tasks where representations that can do pixel-reconstructions aren't what's really required (and they show this experimentally) e.g. distractions exist in images, so they focus on the lossy latents instead. This seems more novel and distinct to simply pre-training some representation using contrastive / masked methods common in the literature. And then in their (extensive) DMC+Atari experiments they augment by removing (pixel) spatial-time cubes in video sequences (given correlations nearby etc) and force the latent structure to capture whatever still needs capturing, perhaps a Q-function input for SAC, trained jointly. So I agree with the other reviewers that this work seems interesting and novel where the NeurIPS community would benefit from reading + understanding it

**Award:**

No

---

### Decision · Program_Chairs · 2022-09-14

Accept